# miR-146 connects stem cell identity with metabolism and pharmacological resistance in breast cancer

Chiara Tordonato[1,2], Matteo Jacopo Marzi[3], Giovanni Giangreco[1,4], Stefano Freddi[1], Paola Bonetti[3], Daniela Tosoni[1], Pier Paolo Di Fiore[1,2]*, and Francesco Nicassio[3]*

**Although ectopic overexpression of miRNAs can influence mammary normal and cancer stem cells (SCs/CSCs), their physiological relevance remains uncertain. Here, we show that miR-146 is relevant for SC/CSC activity. MiR-146a/b expression is high in SCs/CSCs from human/mouse primary mammary tissues, correlates with the basal-like breast cancer subtype, which typically has a high CSC content, and specifically distinguishes cells with SC/CSC identity. Loss of miR-146 reduces SC/CSC self-renewal in vitro and compromises patient-derived xenograft tumor growth in vivo, decreasing the number of tumor-initiating cells, thus supporting its pro-oncogenic function. Transcriptional analysis in mammary SC-like cells revealed that miR-146 has pleiotropic effects, reducing adaptive response mechanisms and activating the exit from quiescent state, through a complex network of finely regulated miRNA targets related to quiescence, transcription, and one-carbon pool metabolism. Consistent with these findings, SCs/CSCs display innate resistance to anti-folate chemotherapies either in vitro or in vivo that can be reversed by miR-146 depletion, unmasking a "hidden vulnerability" exploitable for the development of anti-CSC therapies.**

## Introduction

Cancer stem cells (CSCs) lie at the apex of the hierarchical cellular organization of different types of solid tumors and are thought to drive tumor initiation, therapy resistance, relapse, and metastasis (Al-Hajj et al., 2003; Dalerba et al., 2007; Visvader and Stingl, 2014). There is evidence that the natural history and clinical outcome of cancers are directly related to CSC content. For instance, poorly differentiated breast cancers (BCs), characterized by unfavorable outcome, display a higher CSC content compared with well-differentiated, good-prognosis BCs (Pece et al., 2010), and a transcriptional signature measuring the degree of "stemness" of BCs was shown to be an independent predictor of prognosis (Pece et al., 2019). Moreover, because of their relative quiescent state, CSCs display resistance to conventional anti-cancer therapies, which typically target highly proliferating cancer cells (Creighton et al., 2009; Diehn et al., 2009; Li et al., 2008; Liu and Wicha, 2010).

The emergence of CSCs has been associated with multiple intrinsic (i.e., genetic) and extrinsic cues, leading to different hypotheses about their origin (Visvader and Stingl, 2014). Stem cell (SC) identity is associated with distinctive features connected to the enactment of vast transcriptional and metabolic programs. For instance, the activation of the epithelial-to-mesenchymal (EMT) transcriptional program has frequently been associated with the acquisition of SC properties, and ectopic expression of EMT transcription factors, such as Snail, Twist, and Zeb1/2, has been shown to induce CSC-like phenotypes in vitro and in vivo (Mani et al., 2008; Scheel et al., 2011).

Metabolic reprogramming is also emerging as a key process supporting both normal and cancer SC biology, with particular catabolic and anabolic pathways associated with, and necessary for, the maintenance of an undifferentiated and pluripotent state (Penkert et al., 2016; Shyh-Chang and Ng, 2017). The switch from oxidative phosphorylation to aerobic glycolysis is a common metabolic trait of CSCs, needed to survive under stressful conditions, to fulfill the demand of essential amino acids, nucleotides, and lipids, and to adapt to changes in the tumor microenvironment (Wong et al., 2017).

miRNAs are a class of small noncoding RNAs (18–22 nt) that function in post-transcriptional regulation of gene expression, acting as "sculptors" of the transcriptome and influencing almost every developmental and disease processes (Bartel, 2018). In BC, a number of miRNAs have been linked to inhibition of the

[1]European Institute of Oncology IRCCS, Milan, Italy; [2]Department of Oncology and Hemato-Oncology, Università Degli Studi di Milano, Milano, Italy; [3]Center for Genomic Science of Istituto Italiano di Tecnologia at European School of Molecular Medicine, Istituto Italiano di Tecnologia, Milan, Italy; [4]Tumour Cell Biology Laboratory, The Francis Crick Institute, London, UK.

*P.P. Di Fiore and F. Nicassio contributed equally to this paper; Correspondence to Francesco Nicassio: francesco.nicassio@iit.it; Pier Paolo Di Fiore: pierpaolo.difiore@ieo.it.

CSC phenotype, namely Let-7a, miR-200c, miR-34a, and miR-93 (reviewed in Tordonato et al., 2015). However, their direct involvement in SC/CSC biology is uncertain, as they are poorly expressed or absent in SCs/CSCs. In addition, these miRNAs inhibit SC phenotypes only upon overexpression, through the induction of differentiation and the inhibition of self-renewal determinants (BMI-1 and Notch), transcription factors (ZEB1/2), or signaling pathways involved in EMT (ZEB1/2, MAPK, and STAT3; Aceto et al., 2012; Iliopoulos et al., 2009; Scheel et al., 2011; Shimono et al., 2009; Wellner et al., 2009).

Here, we report that members of the miR-146 family (miR-146a-5p and -146b-5p) are specifically expressed in the SC compartment of the normal mammary gland and in BC cells displaying CSC features. miR-146 controls SC/CSC identity and highlights a metabolic state, likely coopted from normal SCs, that is associated with an intrinsic resistance to anti-cancer drugs, thus providing evidence of a crosstalk between transcriptional and metabolic programs through miRNA activity.

## Results

### Identification and characterization of mammary SC-specific miRNAs

To identify miRNAs differentially expressed in mammary SCs versus progenitors, we employed a previously described FACS-based assay that uses the lipophilic dye PKH26 to isolate highly enriched SC versus progenitor populations from mammospheres (Cicalese et al., 2009; Pece et al., 2010). During mammosphere growth, PKH26 is selectively retained by slowly dividing/quiescent SCs (PKH$^{pos}$), while it is progressively diluted in actively dividing progenitors (PKH$^{neg}$), permitting the separation of these two populations by FACS.

We analyzed miRNA expression (details in Fig. S1, A–C; and Table S1) in PKH$^{pos}$ (SCs) and PKH$^{neg}$ (non-SCs) cells purified from mammospheres generated from (1) primary mouse mammary epithelial cells (MECs), and (2) the human normal MEC line (MCF10A), which contains a SC-like population that is able to differentiate in vitro (Fig. 1 A; Debnath et al., 2003). In these two cell models, we identified three miRNAs commonly regulated in PKH$^{pos}$ cells, defined here as "SC-specific miRNAs": miR-146a/b, miR-331, and let-7a (P value of the overlap, <0.01; Fig. 1 B).

In BC, the proportion of cells with tumor-initiating ability (herein operationally equaled to CSCs) correlates with the molecular/biological characteristics of the tumor and its aggressiveness (Clevers, 2011; Pece et al., 2010; Visvader and Lindeman, 2012). We therefore speculated that the SC-specific miRNAs might be differentially expressed in BCs displaying aggressive features. In the cohort of BC patients from The Cancer Genome Atlas (TCGA; Cancer Genome Atlas Network, 2012), the three miRNAs identified a subgroup of cancers with a SC-like expression pattern (Fig. 1 C). These tumors displayed characteristics of aggressive BC associated with poor prognosis, including (1) a predominant basal-like subtype (Fig. 1 D and Table S2); (2) hormone receptor (estrogen receptor [ER] and progesterone receptor [PgR])–negative status (Fig. 1 E and Table S2); and (3) enrichment of p53 mutations/deletions or Myc

amplification (Fig. 1 E and Table S2; Deming et al., 2000; Green et al., 2016; Miller et al., 2005).

Similar findings were obtained with an independent BC cohort from the Molecular Taxonomy of Breast Cancer International Consortium (METABRIC; Fig. S1, D–F; and Table S3; Curtis et al., 2012), thus confirming that the SC-miRNAs signature can stratify breast tumors according to their biological and molecular features.

### miR-146a/b are enriched in mammary SCs/CSCs versus their progenitors

Of the three identified miRNAs, the relevance of let-7a and miR331-3p to SCs and BC homeostasis has been previously reported (Copley et al., 2013; Leivonen et al., 2014; Yu et al., 2007). We therefore concentrated on miR-146a/b.

Previous studies have shown that mammary SCs and CSCs share common transcriptional traits (Lim et al., 2010; Shipitsin et al., 2007; Visvader and Stingl, 2014). Consistently, our results (Fig. 1, B–E) suggest that the higher levels of expression of miR-146a/b in SCs versus progenitors might be a hallmark of the mammary SC compartment, both in the normal and cancer settings. To investigate this possibility, we initially examined a published list of miRNAs expressed in mammary CSCs, purified using the CD44$^{high}$/CD24$^{low}$ configuration (Shimono et al., 2009). We found that both miR-146a/b were expressed at higher levels in CSCs versus non-CSCs (Fig. 1 F). We purified CSCs from six primary human BC mammosphere cultures using the PKH26 method and observed higher levels of miR-146a/b in PKH$^{pos}$ versus PKH$^{neg}$ cells (Fig. 1 F and Fig. S1 G). Accordingly, we found miR-146a/b up-regulated in a CD44$^{high}$/CD24$^{low}$ subpopulation from the human normal mammary cell line, HMLE, which is enriched in SC-like cells (Fig. 1 F; Al-Hajj et al., 2003; Mani et al., 2008). Thus, miR-146a/b are up-regulated in normal and cancer mammary SCs versus non-SCs, regardless of the methodology used for their purification. miR-146 levels were consistently higher in tumor cell lines (Fig. 1 G) and primary tumors (Fig. S1, H–J) displaying basal-like and mesenchymal features. These tumors exhibit the most aggressive disease course, among the various molecular subtypes of BC, and the highest enrichment in CSCs (Pece et al., 2019). Finally, in the METABRIC dataset, we demonstrated that a high level of miR-146a correlated with reduced overall survival at 20 yr (HR 1.22; P = 0.04; Fig. S1 K).

### Loss of the miR-146 reduces SC/CSC self-renewal in vitro and in vivo

miR-146a/b might simply represent "markers" of the SC state or be directly involved in the specification/maintenance of stemness traits. We investigated these possibilities in primary mouse MECs and SUM159 cells, a BC cell line containing a subpopulation of cells that behaves as CSCs in vitro and in vivo (Fillmore and Kuperwasser, 2008; Gupta et al., 2011; see also Fig. 1 G). In these cells, we silenced miR-146 family expression with a lentiviral sponge (146 kd), which reduced total miR-146 levels by >50% (Fig. 2, A and B) and the levels of miR-146 loaded on RNA-induced silencing complex (where miRNAs function) by >80%, as assessed by Ago2-RNA immunoprecipitation (RIP; Fig. 2 C).

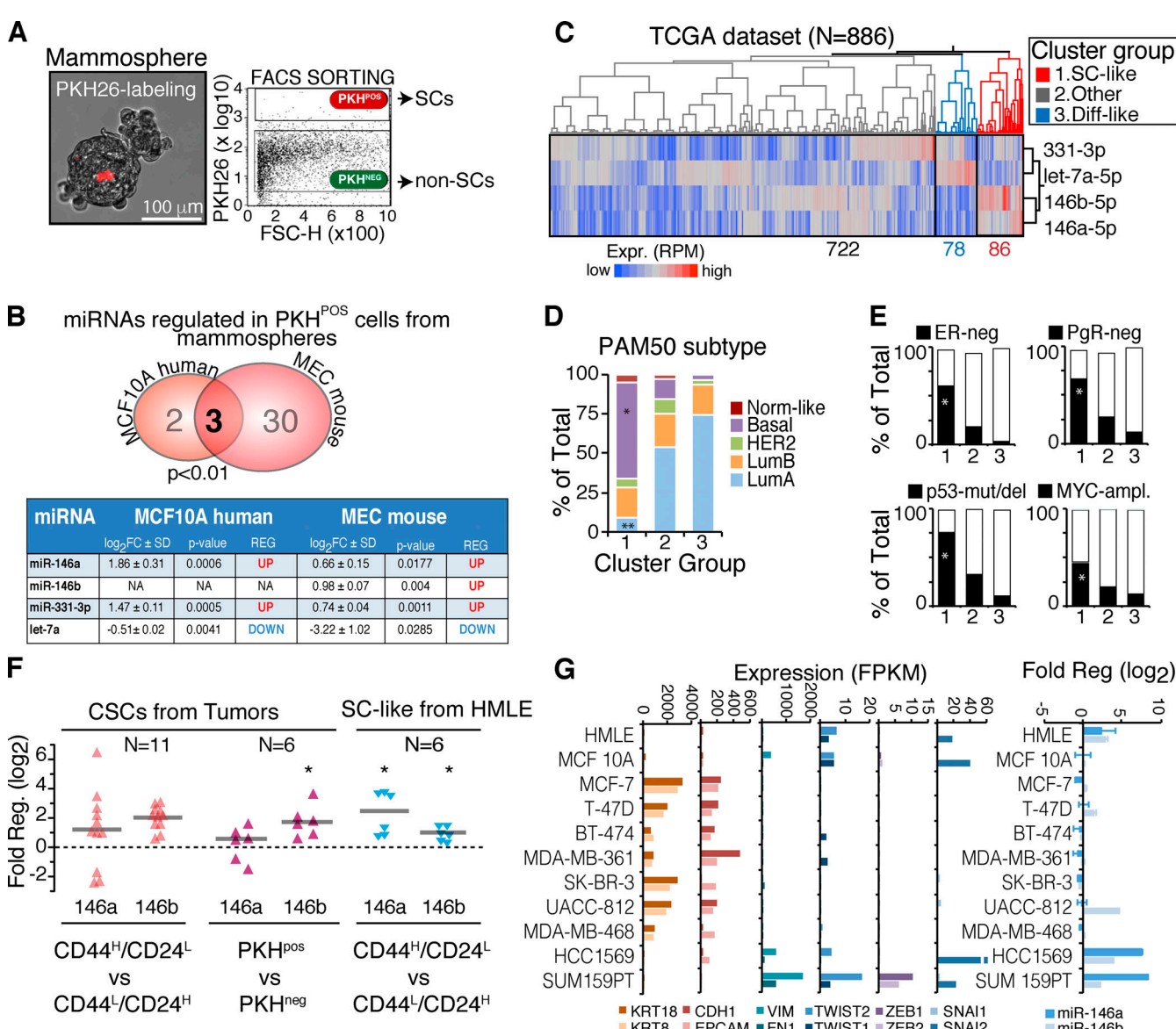

Figure 1. **SC-specific miRNAs and miR-146 expression in SCs and CSCs. (A)** SCs (PKH[pos]) or non-SCs (PKH[neg]) isolated from mammospheres through FACS sorting (n = 2). **(B)** Overlap in differentially regulated miRNAs identified in MCF10A and mouse MECs (P value by Fisher's exact test). "SC-specific miRNAs" (>| 0.5| log₂ fold-change [FC] ± SD, P value <0.05) are in the table**.** Log₂FC is calculated between PKH[pos] versus PKH[neg]. **(C)** Hierarchical clustering (Ward method, with standardized data) of 886 breast primary tumors from TCGA. Three main clusters were identified according to the SC-specific miRNAs. **(D)** Contingency analysis on clusters (identified in C) according to the PAM50 classification. SC-like group (cluster 1) is significantly enriched with basal tumors (*, P < 0.001) and depleted of luminal A tumors (**, P < 0.0001; χ² Pearson coefficient). **(E)** Contingency analysis as in D, according to hormone receptors status (ER and PgR), p53 mutation (mut)/deletion (del) or Myc amplification (ampl.). *, P < 0.0001. **(F)** Regulation of miR-146a/b in CSCs from primary breast tumors isolated with CD44/CD24 markers (left; data are from Shimono et al., 2009), in SC-like cells from the human HMLE (right), or from primary mammospheres with the PKH staining (middle, as in A). *, P < 0.05. **(G)** Left: RNA-seq data (data are from Klijn et al., 2015) reporting the expression of mesenchymal (Vim, FN1, and EMT transcription factors: SNAI1/2, Twist1/2, and ZEB1/2) or epithelial markers (CDH1, Epcam, and KRT8/18) in BC cell lines. Right: Expression by RT-qPCR of miR-146a/b as log₂FC relative to MCF10A, normalized on SNORA73. RPM, reads per million mapped; FSC-H, forward scatter height; Expr., expression; Diff, differentiated-like; NA, not assessed; REG, regulation; FPKM, fragments per kilobase million.

While miR-146 knockdown (KD) had no effect on 2D proliferation (Fig. S2, A–D), it significantly impaired sphere-forming efficiency (SFE; Fig. 2, D and E), suggesting a role of miR-146 in the regulation of self-renewal of normal and cancer mammary SCs. This was assessed directly by limiting dilution transplantation and measuring the frequency of tumor-initiating cells (TICs). We found that the frequency of TICs was significantly reduced in miR-146 KD SUM159 cells as compared with controls

(Fig. 2 F). We next used three BC patient-derived xenografts (PDXs) from a triple-negative subtype, which all maintained the histopathological characteristics of their matched primary tumor (Fig. 3 A and Fig. S2 E) and expressed high levels of miR-146 (Fig. 3 B). The silencing of miR-146 in these PDXs reduced the frequency of TICs by fivefold (Fig. 3, C–F; and Fig. S2 F). When we measured proliferation effects by Ki67 staining (which was possible in two out of three PDXs because of availability of

Figure 2. **miR-146 KD reduces self-renewal in vitro and TIC frequency in vivo. (A and B)** Expression of miR-146 in mammospheres (SUM159 or MECs) infected with CTRL or miR-146 KD lentiviral vector; data as $\log_2$fold relative to CTRL, normalized on SNORD61. **(C)** RIP experiment on SUM159 lysates from CTRL or miR-146 KD cells with Ago2 or IgG antibodies. miRNA expression was analyzed by RT-qPCR. The plot reports the percentage of miRNAs loaded on Ago2 measured as (copies in RIP$_{Ago2\ or\ IgG}$/total copies) * 100. As reference, we used 5% of input ($n$ = 2). **(D and E)** SFE over serial passages of SUM159 (D) or murine MECs mammospheres (E). SFE is reported as a percentage (total number of spheres/total number of cells plated). Data are the mean ± SEM (P value, Student's $t$ test). **(F)** Limiting dilution transplantation of CTRL or miR-146 KD SUM159 cells. TIC frequency calculated using ELDA (Hu and Smyth, 2009); P value calculated by fitting data to the single-hit model.

material), we did not score differences in KD versus SCRAMBLED (SCR; Fig. S2 G). Together, the in vitro and in vivo data support a role for miR-146 in the maintenance of the homeostasis of the SC compartment in the breast gland.

## miR-146 levels stratify cells with SC-like properties

To understand the physiological impact of miR-146 under unperturbed conditions, we generated an miR-146 sensor (Fig. 4, A and B), in which the GFP transgene contained four repeats of a sequence complementary to miR-146a/b in its 3′ UTRs, so that the levels of GFP inversely correlated with miR-146 levels. A second transgene, a truncated form of NGFR (nerve growth factor receptor; ΔNGFR), was used to normalize for lentiviral integrations (Brown et al., 2007).

SUM159 cells, which express high levels of miR-146, showed heterogeneous single-cell expression of miR-146 (Fig. 4, C and D). Thus, we sorted subpopulations with high versus low GFP levels into three different mammary cell types (murine MECs, HMLE, and SUM159) and characterized their biological properties. In all cell types, miR-146[high] cells (GFP[low]) displayed significantly increased SFE versus miR-146[low] cells (GFP[high]; Fig. 4 E–G), suggesting that high miR-146a/b levels distinguish a subpopulation endowed with SC traits. Consistently, purified CD44[high]/CD24[low] SC-like from HMLE cells, which express higher levels of miR-146a/b compared with their CD44[low]/

CD24[high] counterparts (Fig. 1 F), displayed characteristics similar to miR-146[high] cells, including a fibroblast-like appearance (Fig. 4 H), high expression of mesenchymal/SC markers (CD44, CDH2, Snai1, and Serpine1), and low levels of the epithelial/differentiation marker CDH1 (Figs. 4, I and J). Ablation of miR-146 in HMLE cells decreased SC-like properties, including the efficiency of mammosphere formation (Fig. 4 K), whereas the KD in CD44[high]/CD24[low] cells increased the expression of epithelial/differentiation markers (CDH1, MUC1, CD24, and KRT5/18) with concomitant decrease in the mesenchymal marker CDH2 (Fig. 4, L and M). Finally, we isolated CD44[high]CD24[low] cells and followed their reconversion, over time, toward the initial cell heterogeneity in the presence or not of miR-146. As shown in Fig. 4 N, miR-146 KD accelerated the conversion (Mani et al., 2008), confirming that miR-146 is necessary for the maintenance of the mammary SC-like pool.

## miR-146a/b modulates multiple pathways in the mammary SC-like compartment and targets quiescence, DNA, and RNA metabolism genes

To gain mechanistic insights into miR-146 functions in the mammary SC compartment, we performed a comparative analysis of the transcriptomes of: (1) miR-146[low] versus miR-146[high] cells, and (2) miR-146[high] cells in which the expression of miR-146 was ablated (miR-146[high] KD versus SCR; Fig. 5 A).

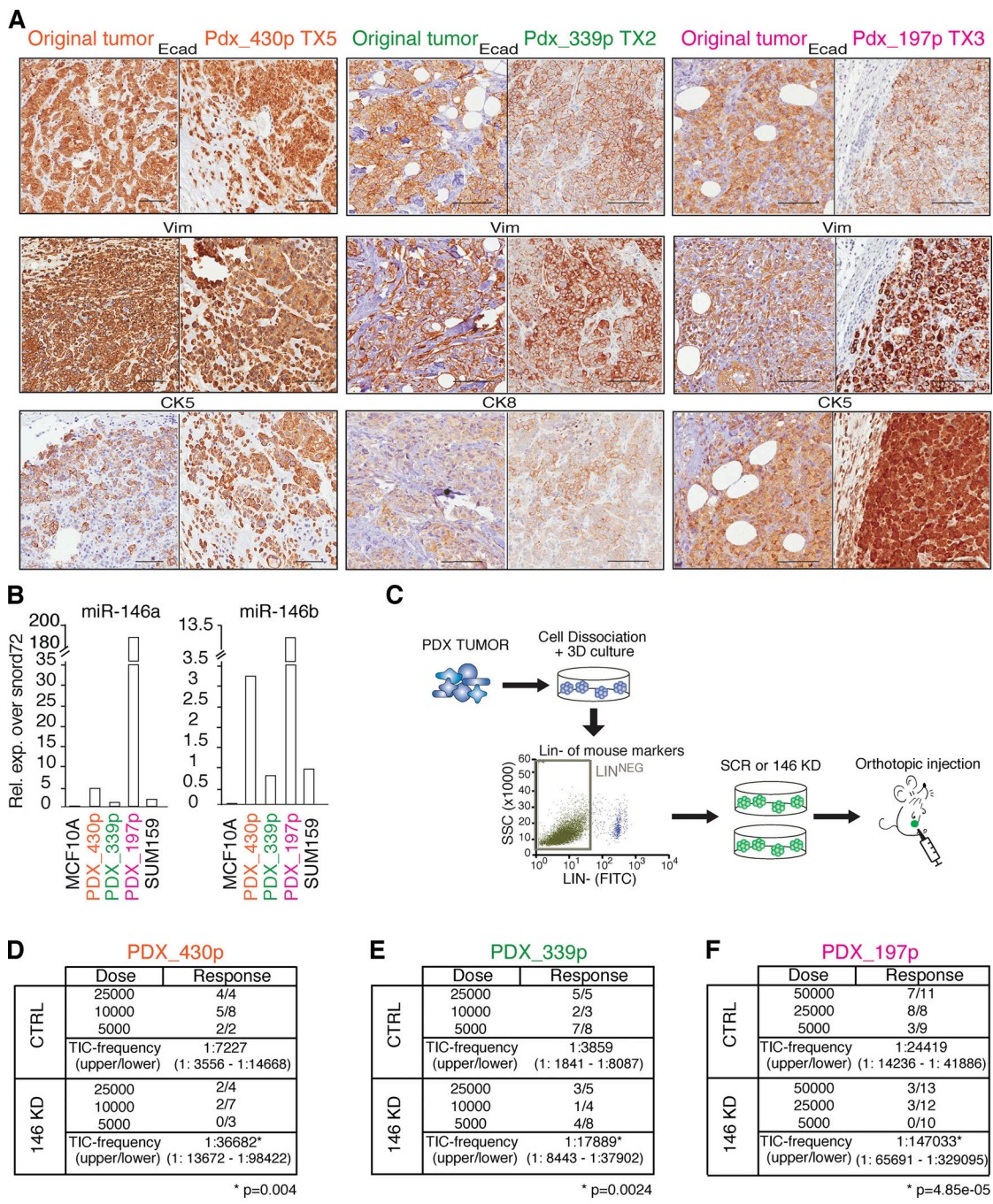

Figure 3. **miR-146 KD reduces TIC frequency in human BC PDXs. (A)** Representative IHC images of PDXs and corresponding original tumors, stained as indicated; scale bar, 100 µm. **(B)** Relative levels of miR-146a/b measured by RT-qPCR in PDXs; SUM159 and MCF10A were used as references. Data normalized on SNORD72. **(C)** Scheme of generation and manipulation of human BC PDXs (see Materials and methods for details). **(D–F)** Limiting dilution transplantation of PDXs 430p, 339p, and 197p expressing CTRL or miR-146 KD vector. TIC frequency calculated using ELDA software (P value calculated by fitting data to the single-hit model). Rel. exp., relative expression; SSC, side scatter; Ecad, E-cadherin; Vim, vimentin; IHC, immunohistochemistry.

This strategy could enrich for direct miR-146 targets, as these transcripts should (1) inversely correlate with miR-146 levels and (2) be induced upon miR-146 KD. We looked for predicted miR-146 targets using TargetScan7.1 (Agarwal et al., 2015) with stringent criteria (context score below −0.15, $n$ = 945). Overall, we found that the predicted miR-146 target genes were slightly (+0.10 median $\log_2$ fold-change) but significantly up-regulated upon either miRNA 146 inhibition (KD) or by comparing cells with different miR-146 levels (146-low versus 146-high), with respect to not-targets (P < 0.001; Fig. 5, B and C.) This magnitude of change is comparable to the change observed in other published work investigating loss of miRNA function (Baek et al., 2008; Wen et al., 2015).

We next investigated whether gene signatures related to mammary SCs correlate with miR-146 levels using gene set enrichment analysis (GSEA). Initially, we used a gene expression

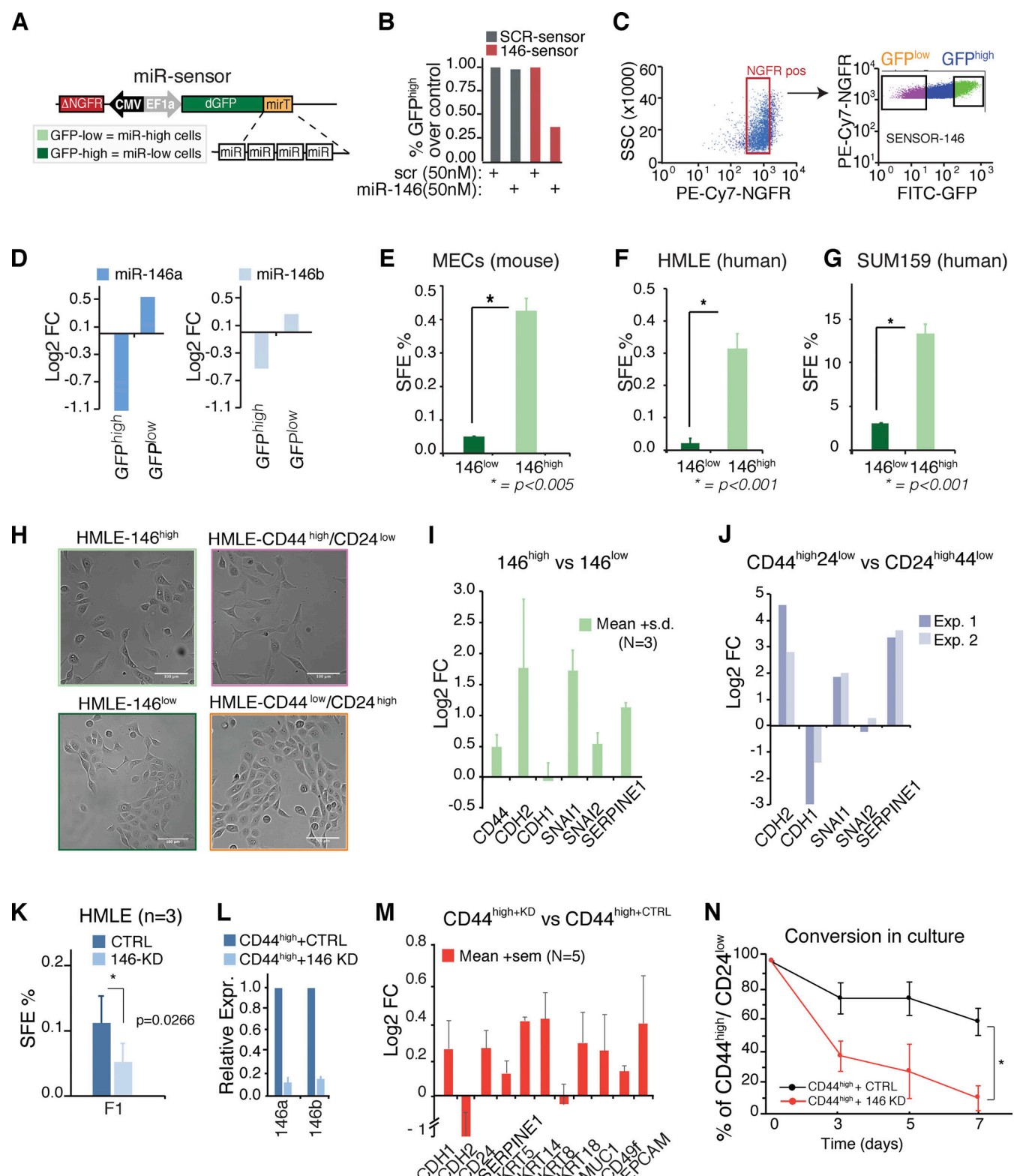

Figure 4. **miR-146 sensor stratifies cells with stem-like properties. (A)** Scheme of the miR-146a/b sensor with four repeats of miR-146 (miRT) in the 3′ UTR of a destabilized GFP (dGFP). **(B)** Percentage of GFP[high] cells upon overexpression of miR-146 or a scrambled control oligo (transfection, 50 nM) in MCF7 cells. Cells were previously transduced with GFP-sensors (146 sensor, responsive to 146 levels, or SCR sensor, used a control). GFP levels were measured by FACS analysis 48 h after transfection and normalized over the SCR-sensor transfected with control. **(C and D)** Biparametric sorting of SUM159 cells transduced with 146-sensor, according to NGFR/GFP levels, used to distinguish ΔNGFR[+] GFP[high] and ΔNGFR[+] GFP[low] cell populations. The levels of miR-146a/b in the two populations are shown in D by RT-qPCR analysis. Data are reported as log[2]fold difference over unsorted cells and normalized to SNORD61. **(E–G)** SFE assay on miR-146[low] versus miR-146[high] subpopulations of primary mouse MECs ($n = 2$), HMLE ($n = 2$), and SUM159 ($n = 3$). Mean ± SEM; P value by Student's $t$ test.

**(H)** Bright-phase images (10×; scale bar, 100 μm) of HMLE subpopulation obtained by FACS sorting according to surface markers or to endogenous miR-146 levels. **(I and J)** Expression of stem-related genes (RT-qPCR) in HMLE subpopulations: miR-146$^{high}$ versus miR-146$^{low}$ (I, mean ± SD) or in CD44$^{high}$/CD24$^{low}$ versus CD44$^{low}$/CD24$^{high}$ (J, two biological replicates). Data are normalized to RPLP0. **(K)** SFE (%) of HMLE cells upon CTRL or miR-146 KD. Data are mean ± SEM; P value by Student's t test. **(L)** Levels of miR-146a/b by RT-qPCR in HMLE CD44$^{high}$/CD24$^{low}$ cells upon CTRL or miR-146 KD. Data are normalized on SNORD61 and over control CD44$^{high}$/CD24$^{low}$ cells ($n$ = 3 ± SD). **(M)** Expression of stem- or EMT-related genes in HMLE CD44$^{high}$/CD24$^{low}$ upon CTRL of miR-146 KD. Data are mean ± SEM, normalized to RPLP0 ($n$ = 5). **(N)** HMLE CD44$^{high}$/CD24$^{low}$ cells were sorted at day 0 (95% of the total) and plated in culture immediately after sorting. Conversion of CD44$^{high}$/CD24$^{low}$ (in CTRL or miR-146 KD cells) toward the original cell population was evaluated after 3, 5, and 7 d, through FACS analysis. The line graph reports the remaining percentage of SC-like cells (CD44$^{high}$/CD24$^{low}$) at each time point. Data are reported as mean ± SEM ($n$ = 2); P value by Student's t test (<0.01); *, P < 0.001. FC, fold-change; Expr., expression; Exp., experiment; SSC, side scatter.

dataset generated in our laboratory by comparing the HMLE SC-like (CD44$^{high}$/CD24$^{low}$) versus differentiated (CD24$^{high}$/CD44$^{low}$) subpopulations and selecting pathways up-regulated or repressed in SC-like cells. Genes up-regulated in SC-like cells (herein referred to as the CD44_UP gene set) were significantly associated with miR-146$^{high}$ cells and down-regulated by miR-146 KD (Fig. 5 D). Conversely, genes down-regulated in SC-like cells (the CD44_DOWN gene set) did not show a coherent reciprocal correlation with miR-146 levels (Fig. S3 A). Using independent mammary SC signatures from normal mouse (Lim et al., 2010; Stingl et al., 2006) or human normal/cancer mammary tissues (Shipitsin et al., 2007), we confirmed that SC-specific gene sets were down-regulated by miR-146 KD and associated with miR-146$^{high}$ cells (Fig. 5 E). In contrast, no coherent correlation was observed for signatures down-regulated in SCs (Fig. 5 E), suggesting that miR-146 expression might be required to sustain genes specifying SC functions, but not directly connected to differentiation. Finally, using the molecular signature database (MSigDB), we noticed that the transition from the miR-146$^{high}$ to the miR-146$^{low}$ state is accompanied by (1) the reduction of several adaptive response pathways involved in mammary SC maintenance as inflammatory (i.e., TNF-α and IFN), p53, hypoxia, and EMT pathways; and (2) the activation of oxidative phosphorylation metabolism, basal transcriptional activity (e.g., Myc targets), and exit from the quiescent state (with activation of the G2–M transition and E2F targets; Fig. S3 B).

To search for direct miR-146–regulated targets in SCs, we adopted a ranking strategy instead of a fold-change cutoff, since transcriptional effects following loss-of-function of miRNAs are typically mild (Baek et al., 2008; Selbach et al., 2008) and may fall out conventional thresholds (see also Fig. 5 B). The four datasets containing miR-146–related gene expression profiles (two datasets each for the 146$^{low}$/146$^{high}$ and KD/SCR comparisons) were ordered from the most up-regulated to the most down-regulated transcript and divided into 10 bins, to select for transcripts with consistent regulation. The first four bins, of all datasets, were significantly enriched in miR-146–predicted targets (Fig. 5 F, highlighted in red); they were therefore merged to select for commonly induced transcripts (1765_UP genes; Fig. 5 G), including 221 predicted direct targets (miR-146 SC targets; Fig. 5 H). Similarly, the last four bins were depleted of predicted direct targets (Fig. 5 F, highlighted in blue) and were merged to obtain commonly repressed genes (1875_DOWN; Fig. 5 G). In the subset of the 1875_DOWN genes, we detected enrichment of pathways connected to BC aggressiveness, stemness, and SC-related properties, such as EMT, inflammatory pathways, or hypoxia (Fig. S3 C and Table S4), suggesting

that miR-146 maintains the SC identity (as observed by GSEA analysis) by indirect transcriptional effects on pathways mostly related to adaptive response mechanisms. Conversely, pathways related to metabolism, RNA transcription, DNA synthesis/repair, and cell cycle/mitosis were enriched among the 1765_UP genes, suggesting a direct role of miR-146 in repressing pathways connected to "exit from quiescence" (Fig. S3 D and Table S4).

When the 221 direct miR-146 targets in SCs were considered alone, they showed a high degree of interconnection, with the most significantly enriched category represented by metabolic processes (one-carbon metabolism, purine synthesis and folate biosynthesis), cell cycle/mitosis, and RNA processing/transcription (Fig. 5 I). Within the one-carbon metabolism category, the direct targets of miR-146 (MTDHF1, MTDHF2, phosphoribosylglycinamide formyltransferase, and dihydrofolate reductase) were also confirmed at the biochemical level by immunoblot analysis on SC-like cells (Fig. S3 E).

## miR-146 role in the determination of resistance to therapy

The sum of the previous data suggests that the loss of miR-146 in SC-like cells has pleiotropic effects, reducing adaptive response mechanisms and activating the exit from quiescent state. These pathways might concur with the determination of the SC state imposed by the expression of miR-146.

Among the many properties of CSC, one that is of particular interest for patients' management is resistance to therapy, a widely reported attribute of CSC (Shibue and Weinberg, 2017) that might be at the basis of therapy failure, especially in the metastatic setting (Oskarsson et al., 2014). We therefore reasoned that drug resistance might represent an exploitable tool to probe into one of the molecular mechanisms through which miR-146 operates, with potential clinical relevance. We employed the SUM159 cell line, which has high CSCs content (SFE ~15–20%; Fig. 2 D) and high miR-146 levels (Fig. 1 G), and exposed it to several chemotherapeutic drugs, under conditions of miR-146 KD. The miR-146 KD induced a modest effect on the drug sensitivity, as measured by IC50 (half-maximal inhibitory concentration), of almost all tested drugs (Fig. 6, A–C; and Fig. S4 A). In sharp contrast, the effect of methotrexate (MTX), which selectively interferes with the folate pathway (Friedman and Cronstein, 2019), was increased by more than 20-fold (Fig. 6 D). Of note, one-carbon pool and folate metabolism emerged as one the main metabolic pathways targeted by miR-146 in SC-like cells in previous analysis.

To extend the validity of the findings, we employed the mammary cell line HMLE, probing the drug sensitivity in the

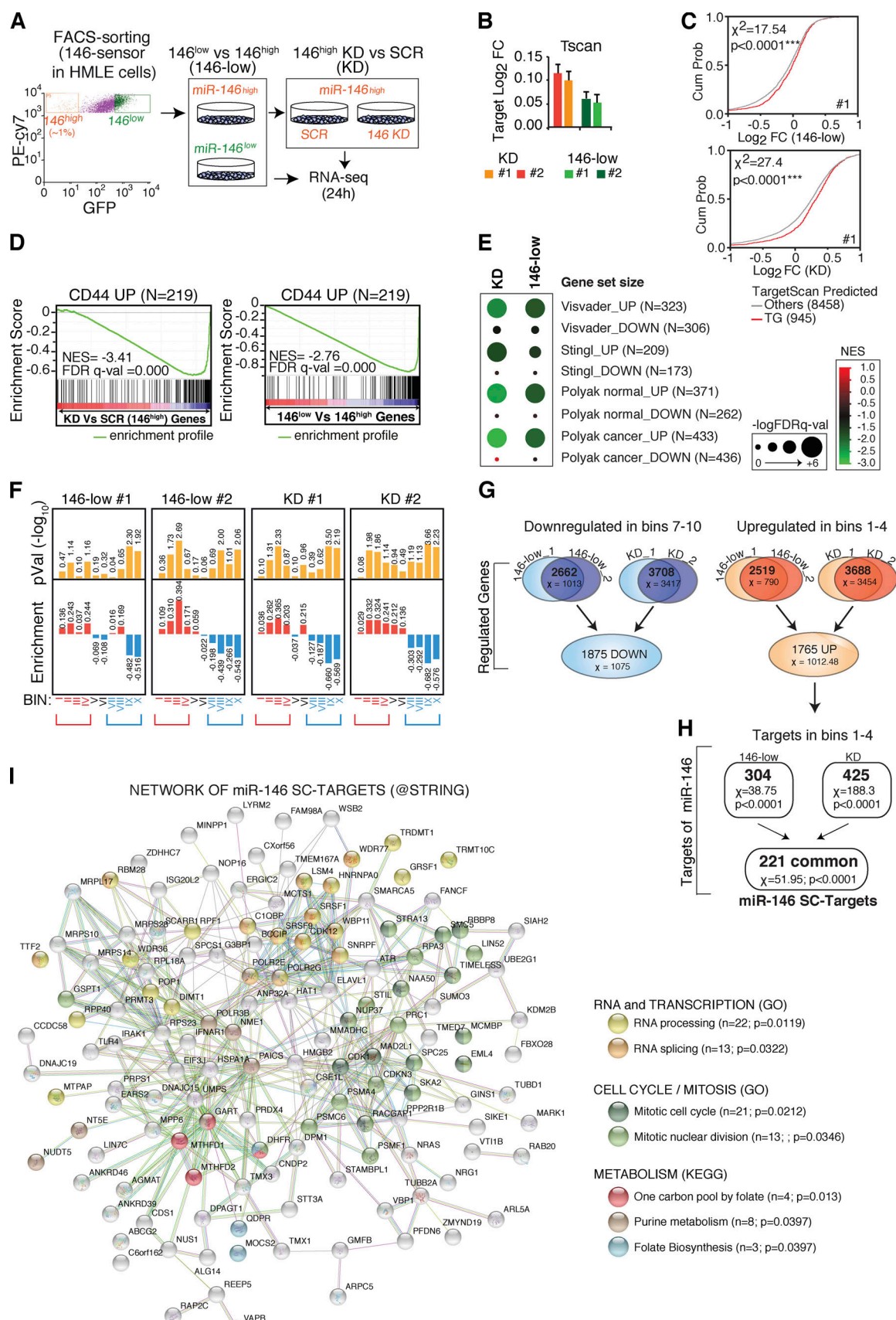

Figure 5. **miR-146a/b targets in SC-like cells. (A)** Strategy for the identification of miR-146 targets in the mammary SCs from HMLE. **(B and C)** The gene regulation of predicted miR-146 conserved targets (n = 945, seed containing targets predicted by TargetScan7.1 [Tscan] with context score less than –0.15) was

assessed upon miR-146 modulation: 146-low (146$^{low}$ versus $^{high}$) or KD (146$^{high+KD}$ versus $^{SCR}$). Shown the median log$_2$ fold-change (FC) ± SEM in each experiment (B), and the cumulative distributions of log$_2$FC (C), compared with predicted nontarget genes (Others, $n$ = 8,458). χ$^2$ and P value by Wilcoxon test. **(D)** GSEA correlating the gene expression of 146$^{low}$ versus $^{high}$ (146-low) or 146$^{high+KD}$ versus $^{SCR}$ (KD) cells with the list of genes up-regulated in CD44$^{high}$ cells (CD44 UP). Significance by normalized enrichment score (NES) and false discovery rate q-value (FDR q-val). **(E)** GSEA analysis (NES and FDR) as in D, with gene lists from published SC signatures (genes up-regulated [UP] or down-regulated [DOWN] in SC cells). **(F)** Ranked gene expression changes in 146-low or KD cells, in two independent experiments (#1, #2), were divided into 10 bins of ≈1,100 genes/bin. Enrichment of predicted targets ($n$ = 945, as in B) was measured in each bin and reported as log$_2$FC of observed versus expected frequency (P value by contingency analysis). Bins 1–4 were enriched for miR-146 predicted targets, while bins 7–10 were depleted. **(G)** Overlap of genes contained in bins 7–10 and 1–4 (as defined in F) between all the experiments. Shown also the significance (χ$^2$, by contingency analysis; P < 0.001 in all cases). 1875 DOWN indicates common down-regulated genes in SCs, while 1765 UP indicates common up-regulated genes in SCs. **(H)** Overlap of miR-146 predicted targets in bins 1–4, defining the set of 221 high-confidence direct targets specific for SCs (χ$^2$ = 51.95, P value < 0.0001). **(I)** miR-146 SC-specific targets ($n$ = 221) were analyzed using STRING software searching for functional association networks. Three highly interconnected categories were identified: RNA/transcription, metabolism, and cell cycle/mitosis (with FDR values). Cum Prob, cumulative probability; TG, targets; GO, gene ontology; KEGG, Kyoto Encyclopedia of Genes and Genomes.

SC-like fraction as compared with the non-SC one. As shown in Fig. 6 E and Fig. S4 B, the SC-like (CD44$^{high}$CD24$^{low}$) subpopulation showed an ∼50-fold reduced sensitivity to MTX versus non-SC cells (CD44$^{low}$CD24$^{high}$) or the bulk population (IC50: 3,560, 71, and 65 nM, respectively). This behavior was dependent on miR-146 levels, since SC-like cells increased sensitivity to MTX (511 nM, 6.9-fold) upon miR-146 KD (Fig. 6 E and Fig. S4 C). Importantly, the non-SC population did not show any change in MTX sensitivity upon miR-146 manipulation (KD or overexpression; Fig. S4 D).

To obtain formal proof for this concept in a tumoral context, we explored the effects on MTX sensitivity upon miR-146 manipulation using in vivo transplantation experiments. SUM159 cells (with or without miR-146 KD) were transplanted orthotopically in the mammary gland of mice by intra-nipple injection and grown either in untreated (saline) or with MTX (60 mg/kg, four cycles) conditions (Fig. 6 F). The KD of miR-146 resulted in slight longer latency in the appearance of palpable tumors versus controls, followed by tumor development with comparable kinetics (Fig. 6 G, left). Though, the number of CSCs, as measured in retransplantation experiments by TIC frequency, was significantly diminished (Fig. 6 H, upper), in agreement with results obtained previously (Fig. 2 D) and in PDXs (Fig. 3, D–F).

The treatment with MTX had a modest effect on tumor growth in control cells, while the combination of miR-146 KD and MTX displayed a potent synergistic effect (Fig. 6 G, right). This interaction was also evident on CSC number, measured in limiting dilution experiments without any additional further treatment. As summarized in Fig. 6 H, MTX had no effects on TIC frequency of control cells, but enhanced significantly the effects of miR-146 KD, with TIC frequency reduced from fourfold to 15-fold as compared with control.

The sum of the above data strongly argues that loss of miR-146 sensitizes tumors to chemotherapy and antifolate treatment in particular, directly targeting the CSC pool.

## Discussion

In this study, we sought to identify miRNA(s) required for the maintenance of the mammary SC phenotype and also "inherited" by the CSC compartment, which could represent potential therapeutic targets in BC. The miR-146 family fulfills these characteristics as (1) they are expressed at high levels in mammary SCs and CSCs versus more differentiated progeny; (2) their

depletion leads to loss of SC features in vitro and in vivo and accelerates the conversion of SCs to non-SCs; and (3) their depletion in SC and CSCs causes increased sensitivity to the chemotherapeutic agent MTX. Thus, the miR-146 family appears to be specifically required to maintain the SC identity in the mammary tissues. In this regard, miR-146s are functionally similar to other SC-specifying miRNAs, such as the miR-290/302 family in embryonic SCs (Wang et al., 2008) or miR-125b in the skin (Zhang et al., 2011).

The role of miR-146 in cancer is, perhaps not surprisingly, rather complex. In BC, previous reports have linked miR-146 expression to the basal subtype (Forloni et al., 2014; Garcia et al., 2011). While we confirmed this association using large BC datasets (TCGA or METABRIC), we have reasons to believe that this is not an intrinsic property of basal BCs per se, but rather a reflection of the high CSC content of these tumors (Pece et al., 2010). Indeed, in every condition that we analyzed, the expression of miR-146 was heterogeneous at the single-cell level and segregated with SC-like phenotypes. In other cancers, miR-146 expression has been reported to be either down- or up-regulated, depending on the context (Testa et al., 2017). However, in most cases, bulk cell populations were analyzed, and some ambiguity in miR-146 expression levels might derive from the presence of nonepithelial contaminants, such as macrophages and regulatory T cells, which are known to express miR-146 at high levels, in particular during inflammatory response (Lu et al., 2010). Thus, it is yet to be established how exactly miR-146 expression is in other epithelial cancers, if it is heterogeneous at the intratumoral level, and, most importantly, whether it demarcates the CSC population. A role for miR-146 in CSCs has been described in colorectal cancer, where it promotes a symmetric mode of division through the Snail/miR-146a/β-catenin/Numb axis (Hwang et al., 2014), and in glioma, where it was shown to inhibit neurosphere formation and tumor development by targeting NOTCH1 (Mei et al., 2011). Therefore, while miR-146 has been linked to CSC behavior in various contexts, the molecular mechanism through which it operates could be rather context-specific.

Different functions, sometimes underlying opposing effects on cancer phenotypes (oncogenic versus tumor-suppressive), have been reported for the miR-146 family, which could be explained by the wide spectrum of miR-146 target genes. With the exception of a few common genes belonging to the inflammatory signaling cascade (e.g., TRAF6 and IRAK1/2), miR-146 targets

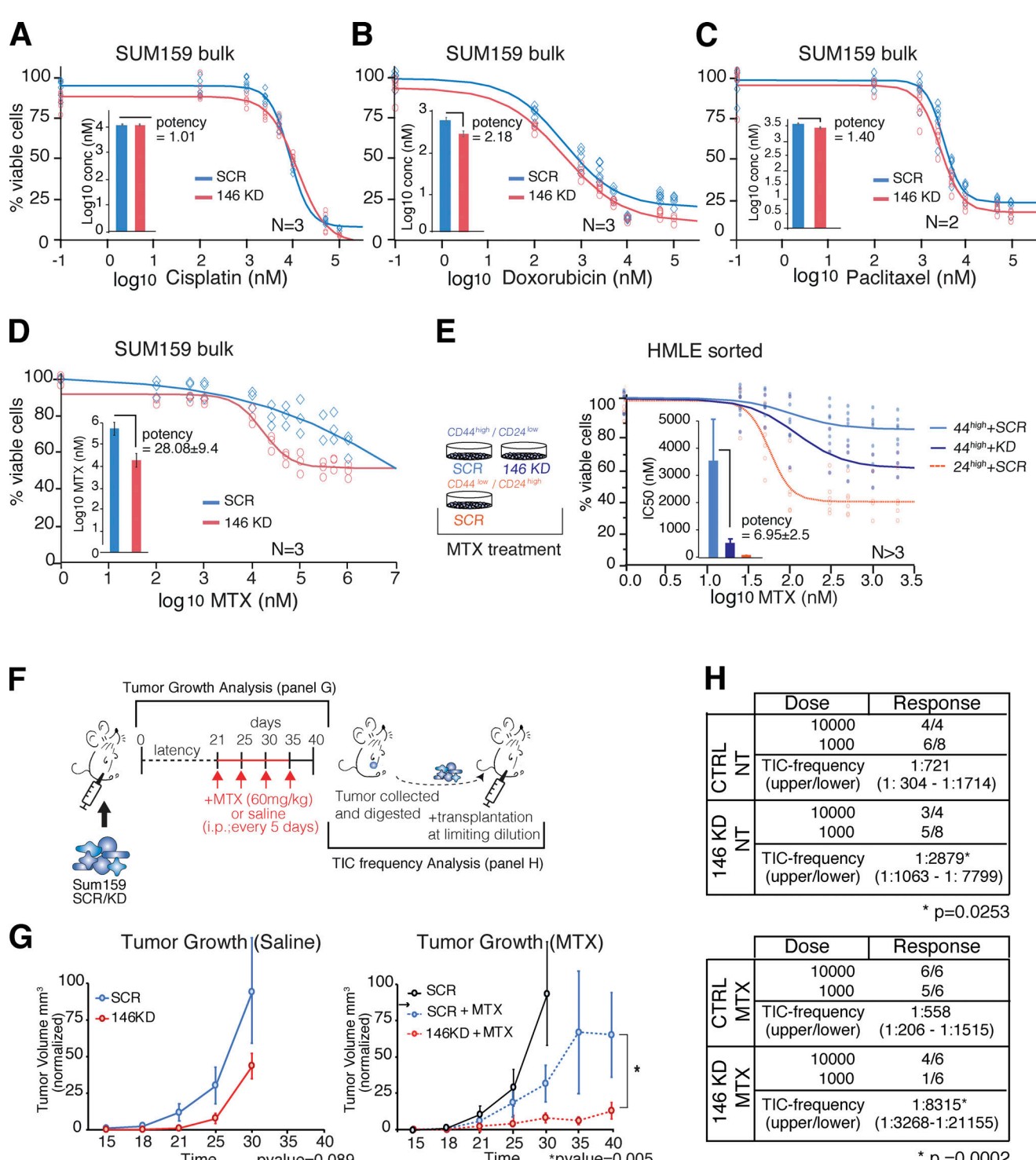

Figure 6. **miR-146 modulates resistance to anti-folate therapy. (A–D)** Dose–response curves of SUM159 CTRL or miR-146 KD cells treated with different chemotherapeutic agents for 72 h. The relative potency of each drug was calculated by fitting the data according to Logistic 4P script (JMP software) over control. The graph bar reports Log10 (drug concentration [conc]) measured as the mean ± SEM of different numbers of biological replicates. **(E)** CD44[high]/CD24[low] and CD44[low]/CD24[high] HMLE subpopulations were FACS sorted, transfected with control or miR-146 KD oligos, and then treated with MTX. Dose–response curves were generated by fitting the data according to Logistic 4P script (JMP software). IC50 is reported in the bar graph as the mean ± SEM of more than three biological replicates. **(F)** SUM159 cells (SCR or KD) were injected into mammary fat pad of NOD.Cg-PrkdcscidIL2rgtm1Wjl/SzJ mice and treated as summarized in the scheme (saline or MTX, 60 mg/kg, dosed every 5 d for four doses). After tumors collection, SCR or KD cells treated or not with antifolate agent were reinjected at limiting dilution. **(G)** Tumor growth and volume were monitored twice a week. Each point of the growth curves represents the tumor volume expressed as the mean value ± SEM ($n$ > 3 tumors for each condition). A paired two-sided Student's $t$ test was used to evaluate the overall tumor volume reduction (*, P < 0.05). In the right plot, the curve for SCR tumors (treated with saline) was duplicated from the left plot (all assays in the right and left panel were performed at the same time). **(H)** Limiting dilution transplantation of SUM159 expressing CTRL or miR-146 KD vector treated (MTX) or not (NT) with antifolate drug. TIC frequency calculated using ELDA software (P value calculated by fitting data to the single-hit model).

appear to be context specific (Taganov et al., 2006). We therefore investigated the genes and the pathways under the control of miR-146 within the mammary SC compartment. We combined the isolation of a miR-146$^{high}$ SC-like population by miR sensor with a loss-of-function approach, to identify broad transcriptional effects of miRNAs under physiological conditions of expression, which might be very different from those observed upon ectopic (and nonphysiological) overexpression. By this approach, we identified a plethora of transcripts that are regulated, directly or indirectly, by miR-146 and might participate in its control over the maintenance of the SC/CSC-like state. Indeed, regulated genes belong to pathways and cellular functions that have been widely linked either to the exit from quiescence (i.e., activation of oxidative phosphorylation metabolism and of the G2–M transition, E2F targets, and cell cycle genes) or to transcriptional programs that promote SC phenotype (inflammatory pathways, hypoxia, and EMT).

The significance and the impact of miR-146 within each of these pathways remain hard to be established, as they are frequently interconnected; however, by narrowing down the candidate list to the set of putative direct miR-146 targets (221 genes), there was significant enrichment of genes acting in a few specific pathways: metabolism, RNA transcription, DNA synthesis/repair, and cell cycle/mitosis.

This allowed us to establish a mechanistic proof of principle linking miR-146 to at least one of the phenotypic properties of CSCs, i.e., drug resistance, in particular resistance to MTX, a chemotherapy agent and immune system suppressant, widely used for the treatment of a variety of cancers, including advanced-stage BC (Yang et al., 2020b). The genetic interaction between miR-146 and MTX has been revealed as highly effective both in vitro and in vivo and further supported by the regulation of the enzymes of the folate biosynthetic pathway by miR-146 observed in the transcriptomic analyses. In addition, drug sensitivity was affected only and specifically in the SC-like population, while no major effects were observed in the non-SC population (Fig. S4 D), which further suggests a context-specific role for miR-146 in the breast SC, rather than a more general effect on bulk epithelial cells.

Our results suggest that miR-146 KD did not induce a general increase in sensitivity to anti-cancer drugs. This would have indicated, in all probability, that drug sensitivity/resistance followed cell identity rather than the specific miR-146–dependent metabolic profile. Rather, loss of expression of miR-146 seemed to confer sensitivity to a specific drug, MTX, as hypothesized based on the transcriptional metabolic pattern; this, in turn, asks questions about the exact mechanism of miR-146–induced MTX resistance.

In this regard, we envision a possible explanation. Folate is the critical cofactor of one-carbon pool metabolism, a process that directly controls nucleotide biosynthesis (purines and pyrimidines), amino acid homeostasis (glycine, serine, and methionine), availability of methyl-groups (methionine/homocysteine), and redox defense (glutathione; Ashkavand et al., 2017; Ducker and Rabinowitz, 2017; Locasale, 2013). In SCs/CSCs, the limited availability of the pathway (determined by the high miR-146 levels) might impose (or contribute to) a quiescent

state, which is a hallmark of the SC-like state. Under these conditions, the cell might be refractory to the inhibition of the pathway, simply because it does not depend on it. Upon exit from the SC-like state and entrance in the transit-amplifying compartment, anabolic cellular demands might require the switch of miR-146 (or the switch of miR-146 might license the cell to fulfill these demands). This situation is mimicked by miR-146 ablation, which we note is indeed accompanied by increased basal transcriptional activity (e.g., Myc targets) and exit from the quiescent state (with activation of the G2–M transition and E2F targets; Fig. S4 B). Under these conditions, the cellular demand and dependency on folate metabolism would represent a "fragility" point that can be unmasked by anti-folate treatment. Alternatively, MTX, which is also known as an anti-inflammatory drug, could cooperate with miR-146 loss in the suppression of the inflammatory (IL1, IL6, and TNF-α) signaling pathway, which is required for sustaining the identity of SCs/CSCs (Yang et al., 2020a). In the context of SCs/CSCs, the optimal output of the inflammatory signaling pathway could be provided through endogenous miR-146, which modulates (and is modulated by) NF-κB activity through a negative feedback loop (Taganov et al., 2006). In the absence of miR-146, NF-κB signaling pathway loses robustness, and thus, SCs/CSCs might become susceptible to the anti-inflammatory action of MTX.

While the molecular validation of these scenarios would require further analysis and metabolic profiling of SC-like versus non-SCs in the presence/absence of miR-146, at the biological level, our results clearly show that interference of miR-146 expression represents an attractive approach to overcome some forms of drug resistance in the clinical settings.

## Materials and methods
### Cell biology procedures and flow cytometry
The SUM159PT cell line (Asterand) was cultured in Ham's F12 medium with 5% fetal bovine North American serum, human insulin (5 μg/ml), hydrocortisone (1 μg/ml), and Hepes (10 mM). The MCF10A cell line (American Type Culture Collection) was cultured in DMEM/F-12 (1:1) with 5% horse serum, hydrocortisone (500 ng/ml), human insulin (10 μg/ml), cholera toxin (100 ng/ml), and human EGF (20 ng/ml). HMLE cells were kindly provided by Robert Weinberg's laboratory (Whitehead Institute for Biomedical Research and Department of Biology, Massachusetts Institute of Technology, Cambridge, MA) and were grown in mamamry epithelial cell growth medium (Lonza) according to the manufacturer's protocol. Mammary glands from 5-wk-old FVB/Hsd females (Harlan Laboratories) were established as described previously (Cicalese et al., 2009). Briefly, glands were mechanically and enzymatically digested in EDM medium: DMEM plus Ham's F12 (1:1) medium supplemented with human insulin (1 μg/ml,) hydrocortisone (0.5 μg/ml), human EGF (20 ng/ml), 200 U/ml collagenase type 1A (Sigma-Aldrich), and 100 U/ml hyaluronidase (Sigma-Aldrich) for 3 h at 37°C. After digestion, cell suspension was filtered through 100-, 70-, 40-, and 20-μm filters, and red blood cells were lysed using ammonium-chloride-potassium lysis buffer. MECs from primary tumors were collected at the European

Institute of Oncology (Milan, Italy) from patients who had given the informed consent to use of biological materials for scientific purposes. Primary tissues were digested as described in Dontu et al. (2003). All the cells were grown in a humidified atmosphere at 5% $CO_2$ at 37°C, except for SUM159PT, which were grown at 10% $CO_2$.

For mammosphere culture, SUM159, MCF10A, and HMLE cells were plated in ultra-low attachment dishes (Falcon) coated with Poly(2-hydroxyethyl methacrylate) (Sigma-Aldrich) at a density of 1,000 cells/ml in serum-free mammary epithelial medium (Lonza) supplemented with 5 µg/ml insulin, 0.5 µg/ml hydrocortisone, 2% B27 (Invitrogen), 20 ng/ml EGF, 20 ng/ml human b-FGF, and 4 µg/ml heparin. Mammosphere cultures of human and mouse primary samples were plated at a density of 5,000 cells/ml. For serial propagation, mammospheres were collected after 7 d of culture, enzymatically dissociated with trypsin-EDTA (0.025%), and plated at the same density for successive generations. PKH26 (Sigma-Aldrich) staining was performed on MCF10A and primary tissues as described in Cicalese et al. (2009) and Pece et al. (2010; Fig. S1 G). PKH-labeled mammospheres were collected after 7–8 d and enzymatically dissociated with trypsin-EDTA to a single-cell suspension. Each human PKH-labeled mammosphere preparation was depleted of contaminants with CD31 and CD45 microbeads (MACS technology) and subsequently stained with DAPI (1 mg/ml, diluted 1:200 in PBS) for 1 min at RT to select for living cells. Finally, cells were FACS sorted to collect PKH[pos] and PKH[neg] cells in 96-well plates.

Cells infected with miR-146 sensors were blocked with PBS 10% BSA for 10 min at 4°C and stained with anti-ΔNGFR/PE-cy7 (CD271-PeCy7; clone C40-1457; BD Pharmigen) for 15 min at 4°C. Two-color flow cytometry (GFP and PE-cy7) was used to collect ΔNGFR[+]-GFP[high] and/or ΔNGFR[+]-GFP[low] populations. To FACS sort CD44[high]CD24[low] and CD44[low]CD24[high] populations from HMLE, cells were blocked with PBS-BSA 10% for 1 h at 4°C and then stained with CD44-APC (clone C26; BD Pharmigen) and CD24-PE (clone ML5; BD Pharmigen) antibodies for 45 min at 4°C.

### Lentiviral constructs and viral infection
Lentiviral backbone (Bd.LV.miRT vector) for miR-146 sensor was courtesy of L. Naldini (San Raffaele Telethon Institute for Gene Therapy, San Raffaele Scientific Institute, Milan, Italy) and modified as follows: two DNA sequences containing four miRNA response elements with perfect complementarity to miR-146a/b at the 3′ UTR of GFP (miRT) were synthesized (Primm) as follows: sensor 146 sense: 5′ → 3′, CTAGAAAGCCTATGGAATTCA GTTCTCACGATAAGCCTATGGAATTCAGTTCTCAACCGGTAA GCCTATGGAATTCAGTTCTCATCACAAGCCTATGGAATTCAG TTCTCAC; sensor 146b antisense: 5′ → 3′, CCGGGTGAGAACTGA ATTCCATAGGCTTGTGATGAGAACTGAATTCCATAGGCTTAC CGGTTGAGAACTGAATTCCATAGGCTTATCGTGAGAACTGAA TTCCATAGGCTT.

1 µl of each oligo (100 µM) was annealed in a final volume of 50 µl Annealing Buffer (Promega) for 4 min at 95°C, then 10 min at 70°C. Diluted annealed oligos (1:10) were ligated with 100 ng of lentiviral backbone (Bd.LV.miRT vector) doubled-digested

with XhoI and XbaI. Ligation protocol was performed with Quick T4 DNA Ligase according to the manufacturer's indications (New England Biolabs). After cloning, each positive clone sequence was verified by DNA sequencing. miR-146 KD and nontargeting scramble control (CTRL) were commercially available vectors (pmiRZIP lentivector) from System Bioscience (clone MZIP000-PA-1 for CTRL and MZIP146b5p-PA-1 for miR-146 KD).

For virus packaging, pRSV-Rev, pMDLg/pRRE (gag&pol), pMD2.G (VSV-G), and lentiviral vectors (pmiRZIP lentivector or miR-sensor) were cotransfected in HEK293T cells via the calcium phosphate method. The viral supernatant was collected at 36 h after transfection, filtered with a 0.22-µm syringe filter, and ultra-centrifuged for 2 h at 19,800 rpm at 4°C. The viral pellet was then resuspended in mammary epithelial medium at 100× concentration. Viral stock was frozen (–80°C) or directly used to infect target cells in the presence of 1 µg/ml of polybrene. Cells infected were then selected with puromycin for 2–3 d to select stable clones.

### miR-146 overexpression and KD
For miR-146 overexpression, cells were transfected with Hi-PerFect (Qiagen) according to the fast-forward protocol with miRNA Mimic (we used for miR-146 overexpression mimic MSY0002809; and for control, the all-star negative control siRNA SI03650318; Qiagen) at a final concentration of 50 nM. For miR-146 KD, cells were transfected with HiPerFect (Qiagen) according to the fast-forward protocol with the miRNA power family inhibitor at a final concentration of 100 nM (hsa-miR-146 miRCURY LNA microRNA Power family inhibitor; and as control, Negative Control A; Exiqon).

### Cell viability analysis
SUM159 cells infected with miR-146 KD lentivirus were plated in 96-well plates (5,000 cells/well) and treated with drugs at different concentrations for 72 h. HMLE CD44[high]24[low] or CD44[low]24[high] were plated in 12-well plates, then transfected with anti-miR146 or CTRL oligos. 24 h after transfection, cells were treated with MTX for 72 h. Viability was assessed using the Cell-Counting Kit-8 viability kit (CCK-8; Dojindo) according to the manufacturer's protocol.

### Serial transplantation of human PDXs and in vivo experiments
Immunodeficient NOD.Cg-PrkdcscidIL2rgtm1Wjl/SzJ mice were anesthetized by intraperitoneal injections of 150 mg/kg of tri-bromoethanol (Avertin), and fresh specimens from human primary tumors were implanted in the fourth inguinal mammary gland of 4–5-wk-old animals. Mice were monitored twice weekly by an investigator and were euthanized after 3–5 mo when the tumors were ~0.5–1 cm in the largest diameter (depending on the intrinsic variability of human specimens). Human PDXs were collected and mechanically/enzymatically digested in EDM medium for 4 h at 37°C. Cell suspensions were filtered through 100-, 70-, 40-, and 20-µm filters, and red blood cells were lysed with ammonium-chloride-potassium lysis buffer. After 24 h in mammosphere culture, cells were cleaned of murine contaminants with the mouse epithelial cell enrichment kit

(StemCell Technologies) and the dead cells removal kit (Miltenyi Biotec). Pure human epithelial populations were then infected with CTRL or miR-146 KD lentivirus and puromycin-selected. SUM159 were infected with CTRL or miR-146 KD lentivirus and puromycin-selected before injection. For in vivo limiting dilution transplantation experiments, decreasing concentrations of infected cells (SUM159 or human PDXs) were resuspended in a mix of 14 µl PBS and 6 µl Matrigel and transplanted via intra-nipple injection in the fourth inguinal mammary gland of 6–8-wk-old animals. Animals were euthanized after 1–5 mo (depending on tumor latency) when the tumors were ∼0.5–1.2 cm in the largest diameter. Transplantation frequency was calculated with the Extreme Limiting Dilution Analysis (ELDA) web tool, and single-hit assumption was tested for each experimental setting. For therapeutic treatment with MTX, SUM159 tumors with miR-146 KD or CTRL were monitored until they reached a mean volume of 4–6 mm³. Tumor volume was calculated according to the formula ($L * W^2/2$), where L is the length of the longest diameter and W is the length of the shorter diameter. Mice were randomly assigned to different groups (treated or saline) with a minimum of three to five animals/group. Animals received intraperitoneal injection of vehicle drug (saline) or MTX at 60 mg/kg dosed every 5 d, for a total of four injections. Changes in tumor burden were assessed every 3 d with calipers. Animals were euthanized after 40 d when the tumors reached ∼1.2 cm in the largest diameter. Tumors, treated or not with MTX, were digested as previously described. After 24 h in mammosphere culture, we checked for GFP expression, we removed murine contaminants and dead cells, and then we reinjected the cells (SCR ± MTX or KD ± MTX) at limiting dilutions. All animal studies were conducted with the approval of the Italian Minister of Health (762/2015-PR) and were performed in accordance with Italian law (D.lgs. 26/2014), which enforces Directive 2010/63/EU of the European Parliament and of the Council of September 22, 2010, on the protection of animals used for scientific purposes.

## Immunoblotting and immunohistochemistry

Cell lysates were extracted with RIPA lysis buffer (50 mM Tris-HCl, 150 mM NaCl, 1 mM EDTA, 1% Triton X-100, 1% sodium deoxycholate, and 0.1% SDS), supplemented with a protease inhibitor cocktail (Calbiochem) and phosphatase inhibitors. Lysates were clarified by centrifugation at 16,000 $g$ for 10 min at 4°C, and protein concentration was measured by the Bradford assay (Bio-Rad) according to the manufacturer's instructions. Proteins were resolved in 4–20% Protean TGX Precast gel (Bio-Rad), then transferred to nitrocellulose filters. Filters were blocked in 5% milk in TBS 0.1% Tween. After blocking, filters were incubated with the following primary antibodies: phosphoribosylglycinamide formyltransferase (4D6-1D5; NovusBio), dihydrofolate reductase (EPR5284; Abcam), MTHFD1 (C-3; Santa Cruz), and MTHFD2 (D8W9U; Cell Signaling). As a normalizer, we used γ-tubulin (homemade clone).

Filters were finally incubated with the appropriate HRPconjugated secondary antibody (anti-mouse IgG HRP-linked 7076 or anti-rabbit IgG HRP-linked 7074; Cell Signaling) diluted 1:2,000 in TBS 0.1% Tween for 30 min. The signal was revealed using the ECL method (Amersham) with Image Lab software (v3.0; Bio-Rad).

Paraffin sections were twice deparaffinized with Bio Clear (Bio-Optica) for 15 min and hydrated through graded alcohol series (100%, 95%, and 70% ethanol and water) for 5 min. Antigen unmasking was performed with 0.1 mM citrate buffer (pH 6) or EDTA (pH 8) for 50 min at 95°C. Slides were cooled for 20 min at RT then washed in water and treated with 3% $H_2O_2$ for 5 min at RT. Then, slides were preincubated with an antibody mixture (2% BSA, 2% normal goat serum, 0.02% Tween 20 in TBS) for 20 min at RT and stained with primary antibody for 1 h at 37°C. As primary antibodies, we used rabbit anti-human estrogen receptor (1:40; Dko), mouse anti-hKi67 (1:200; Dko), mouse anti-cytokeratin 5 (1:200; Abcam), mouse anti-cytokeratin 8 (1:10; Abcam), and mouse anti-vimentin (1:50; Dko). Slides were then incubated with a secondary antibody (DAKO Envision system HRP rabbit or mouse) for 30 min at RT and finally incubated in peroxidase substrate solution (DAB DAKO) for 2–10 min. Stained slides were digitalized at 20× magnification using the Aperio Scanscope XT (Leica Biosystems) and acquired with the Aperio ImageScope software (Leica Biosystems).

## Total RNA extraction and RT quantitative PCR (qPCR)

Cells were lysed in TRIzol reagent (Invitrogen), and total RNA was extracted with miRNeasy Mini columns or miRNeasy micro-columns according to the manufacturer's protocol. miR-NAs were reverse-transcribed with an miScript II reverse transcription kit (Qiagen), and mature miRNAs were detected with miR-146a (MS00003535) and miR-146b-5p (MS00003542) assays from Qiagen. As controls, we used SNORD61 (MS00033705) or SNORD72 (MS00033719). For gene detection, total RNA was reverse-transcribed with the SuperScript VILO cDNA Synthesis Kit (Life Technologies), and genes were analyzed with Quantifast SYBR green master mix (Qiagen) or SsoFast supermix (Bio-Rad). The complete list of primers used in this study is as follows: CD44: (forward) 5′-ATAGCACCTTGCCCACAATG-3′, (reverse) 5′-TTGCTGCACAGATGGAGTTG-3′; CD24: (forward) 5′-TCAGGCCAAGAAACGTCTTC,-3′, (reverse) 5′-TCCTTGCCA CATTGGACTTC-3′; CDH1: (forward) 5′-TGCCCAGAAAATGAA AAAGG-3′, (reverse) 5′-GTGTATGTGGCAATGCGTTC-3′; CDH2: (forward) 5′-ACAGTGGCCACCTACAAAGG-3′, (reverse) 5′-CCGAGATGGGGTTGATAATG-3′; RPLP0: (forward) 5′-TTCATT GTGGGAGCAGAC-3′, (reverse) 5′-CAGCAGTTTCTCCAGAGC-3′; ACTB: (forward) 5′-TCTACAATGAGCTGCGTGTG-3′, (reverse) 5′-TGGATAGCAACGTACATGGC-3′; SNAI1: (forward) 5′-GGT TCTTCTGCGCTACTGCT-3′, (reverse) 5′-TAGGGCTGCTGGAAG GTAAA-3′; SNAI2: (forward) 5′-ACGCCTCCAAAAAGCCAAAC-3′, (reverse) 5′-ACACAGTGATGGGGCTGTATG-3′; SERPINE1: (forward) 5′-AAGACTCCCTTCCCCGACTC-3′, (reverse) 5′-CAG TGCTGCCGTCTGATTTGT-3′; MUC1: (forward) 5′-TGCCGCCGA AAGAACTACG-3′, (reverse) 5′-TGGGGTACTCGCTCATAGGAT-3′; KRT5: (forward) 5′-AGGAGTTGGACCAGTCAACAT-3′, (reverse) 5′-TGGAGTAGTAGCTTCCACTGC-3′; KRT14: (forward) 5′-TGAGCCGCATTCTGAACGAG-3′, (reverse) 5′-GATGACTGC GATCCAGAGGA-3′; KRT8: (forward) 5′-CAGAAGTCCTACAAG GTGTCCA-3′, (reverse) 5′-CTCTGGTTGACCGTAACTGCG-3′;

**Table 1.**   **Applied Biosystems' HT qPCR profiling protocol as improved by the authors**

| Step or dilution | LSI (µl) | 10 ng Applied (µl) |
|---|---|---|
| RNA to RT | 3 µl | 10 ng |
| RT (two reactions) | 7.5 | 7.5 |
| Pooled RT reaction | 15 | 15 |
| RT to PreAmp (preamplification) | 5 | 5 |
| Final PreAmp | 25 | 25 |
| PreAmp cycles | 16 | 14 |
| Post-PreAmp dilution | 1:4 | 1:4 |
| Dilution PCR | 1:20 | 1:50 |
| Final dilution factor | 1:80 | 1:200 |

KRT18: (forward) 5′-TCGCAAATACTGTGGACAATGC-3′, (reverse) 5′-GCAGTCGTGTGATATTGGTGT-3′; CD49f: (forward) 5′-ATGCACGCGGATCGAGTTT-3′, (reverse) 5′-TTCCTGCTT CGTATTAACATGCT-3′; EPCAM: QT00000371 (Qiagen); mKI67: HS01032443_m1 (TaqMan); and RPLP0_TaqMan: (forward) 5′-CCATTGAAATCCTGAGTGATGTG-3′, (reverse) 5′-TCGCTGGCT CCCACTTTG-3′.

## miRNAs high-throughput (HT) profile and low sample input (LSI) protocol

For the analysis of PKH$^{pos}$ and PKH$^{neg}$ cells isolated from murine primary MECs, we reverse-transcribed total RNAs with Megaplex RT Primers mix and amplified with Megaplex PreAmp Primers (rodent pool A). For miRNA HT profiling, we used the TaqMan Low Density Array Rodent V2.0 (Applied Biosystems), following the manufacturer's instructions. For the analysis of PKH$^{pos}$ and PKH$^{neg}$ cells from MCF10A mammospheres, we isolated 40 PKH$^{pos}$ and PKH$^{neg}$ cells, lysed directly in single-cell Lysis Buffer (Ambion). Total RNA was reverse-transcribed with Human Megaplex RT Primers mix and amplified with Human Megaplex PreAmp Primers (pool A). For miRNA HT profiling, we used the TaqMan Low Density Array Human V2.1 (Applied Biosystems; Table S4).

For the LSI setup, we collected by FACS sorting no more than 200 PKH$^{pos}$ and 200 PKH$^{neg}$ cells in 96-well plates in 10 µl of Single Cell Lysis Kit plus DNase (Ambion). Total RNAs were reverse-transcribed using Human Megaplex RT Primers mix, followed by preamplification with Human Megaplex PreAmp Primers (pool A). Then HT qPCR profiling was performed on TaqMan human platform A V2.1 (Applied Biosystems). We improved the original protocol from Applied Biosystems (Table 1).Raw data (i.e., cycle threshold [Ct] values) were exported to Excel (Microsoft). miRNAs with raw Ct >28 or not expressed (e.g., not amplified) were excluded from the analysis. Expressed miRNAs (Ct <28) were then normalized over the median of housekeeping controls (RNU44, RNU48, and RNU6B) for human array and over the median of U6b, SnoRNA135, and SnoRNA202 for rodent array. Regulated miRNAs were selected based on the following criteria: P value < 0.05, |log$_2$ fold| > 0.5.

## Ago2 RIP

The Ago2 RIP experiment was performed using the Imprint RNA Immunoprecipitation kit (Sigma-Aldrich). Briefly, 10$^7$ cells were lysed in mild lysis buffer (plus Protease Inhibitor Cocktail and RNase inhibitor) for 15 min on ice. Then the lysate was pelleted at 16,000 $g$, 4°C, for 10 min. A fraction (5%) of supernatant was collected as input for RNA control. For each RIP, protein A magnetic beads were preloaded with 2.5 µg Ago2 antibody (rat monoclonal; clone 11A9; Sigma-Aldrich) or 2.5 µg of IgGs from rat serum, at RT for 30 min with rotation. RNAs were immunoprecipitated with Ago2 antibodies or rat IgGs overnight at 4°C with rotation. RIPs were then washed, and RNA was purified with TRIzol LS reagent (Life Technologies) plus miRneasy Micro kit and analyzed with RT-qPCR.

## RNA sequencing (RNA-seq) and GSEA analysis

Total RNA was extracted with the miRNeasy Micro kit (Qiagen) and treated on-column with DNase (Qiagen). Then 500 ng was purified with the Ribozero rRNA removal kit (Illumina). Libraries were generated with the TruSeq RNA Library Prep Kit v2 (Illumina). Next, sequencing was performed on Illumina HiSeq 2000 at 50-bp single-read mode and 50 million reads depth. RNA-seq Next Generation Sequencing reads were aligned to the human hg38 gencode v25 reference genome using the TopHat aligner (version 2.0.6) with default parameters. Differentially expressed genes were identified using the Bioconductor package DESeq2 based on read counts, considering genes whose q value relative to the control is lower than 0.05 and whose maximum expression is higher than reads per kilobase of exon per million mapped reads of 1.

GSEA (http://www.broadinstitute.org/gsea/index.jsp) was performed using the 11,000 genes expressed in HMLE cells obtained from RNA-sequencing in Fig. 5. As gene sets to calculate the normalized enrichment scores, we used four SC signatures (CD44$^{high}$, Polyak, Stingl, and Visvader) subdivided in STEM_UP and STEM_DOWN genes. P values were calculated by performing 1,000 random permutations of gene labels to create ES-null distribution.

## Data availability

The RNA-seq dataset for this study has been deposited in GEO under accession no. GSE131876.

## Statistics

All the analyses (Oneway, Contingency, Principal Component Analysis, IC50 calculation) and statistics related were produced using JMP 12 (SAS) software. Microsoft Excel was used to generate bar graphs with average and SD of repeated experiments, with number of replicates and the statistical test indicated in figure legends. Hierarchical clustering was generated by Cluster 3.0 software (C Clustering Library 1.53) and heatmaps by Java Tree-View software (http://jtreeview.sourceforge.net) for Mac OSX.

## Clinical samples

Fresh or archival formalin-fixed paraffin-embedded mammary primary specimens were collected at the European Institute of Oncology, via standard operating procedures approved by the Institutional Ethical Board. Only samples for which patients gave informed consent were used in the present study.

## Online supplemental material

Fig. S1 shows generation of the LSI protocol and analysis of miR-146 levels in BC datasets. Fig. S2 shows characterization of human BC PDXs and effects of miR-146 KD on proliferation. Fig. S3 shows analysis of miR-146 transcriptional effects. Fig. S4 shows sensitivity to chemotherapy treatment upon miR-146 manipulation. Table S1 shows raw data from the LSI protocol. Table S2 shows analysis of SC-specific miRNAs' signature in the TCGA dataset. Table S3 shows analysis of SC-specific miRNAs' signature in the METABRIC dataset. Table S4 shows a list of pathways regulated by miR-146 in SC-like cells.

## Acknowledgments

We thank first all the patients who donated their biopsy specimens for research purposes; the Genomic Unit at the Italian Institute of technology for sequencing runs; the European Institute of Oncology (IEO) Imaging Unit for FACS-sorting experiments; the Veterinary Facility at FIRC Institute of Molecular Oncology; C. Luise, G. Jodice, and G. Bertalot at the IEO Molecular Pathology Unit for the immunohistochemistry analyses; M. Coazzoli for technical assistance with in vivo experiments; S. Confalonieri for the survival analysis on the METABRIC dataset; the IEO Pharmacy for providing drugs; and R. Gunby for critically editing the manuscript.

This work was supported by grants from the Associazione Italiana per la Ricerca sul Cancro (MCO10000, IG18988, and IG23060 to P.P. Di Fiore; IG14085, IG18774, and IG22851 to F. Nicassio), the Fondazione Italiana per la Ricerca sul Cancro triennial fellowship "Livia Perotti" (project code 18224 to C. Tordonato), the Associazione Italiana per la Ricerca sul Cancro fellowship "Isabella Gallo" (project code 22386 to G. Giangreco), the Fondazione Cariplo (2015-0590 to F. Nicassio), the Italian Ministry of University and Scientific Research (to P.P. Di Fiore), and the Italian Ministry of Health (RF-2016-02361540 to P.P. Di Fiore). This work was also partially supported by the Italian Ministry of Health with Ricerca Corrente and 5x1000 funds.

The authors declare no competing financial interests.

Author contributions: Experiments were performed by C. Tordonato; M.J. Marzi performed RNA-seq analyses; S. Freddi and C. Tordonato performed sorting and imaging; D. Tosoni initially established human PDXs; and P. Bonetti and G. Giangreco helped with mouse experiments. C. Tordonato, F. Nicassio, and P.P. Di Fiore planned the experiments, analyzed the results, and wrote the manuscript.

Submitted: 10 September 2020

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

# Supplemental material

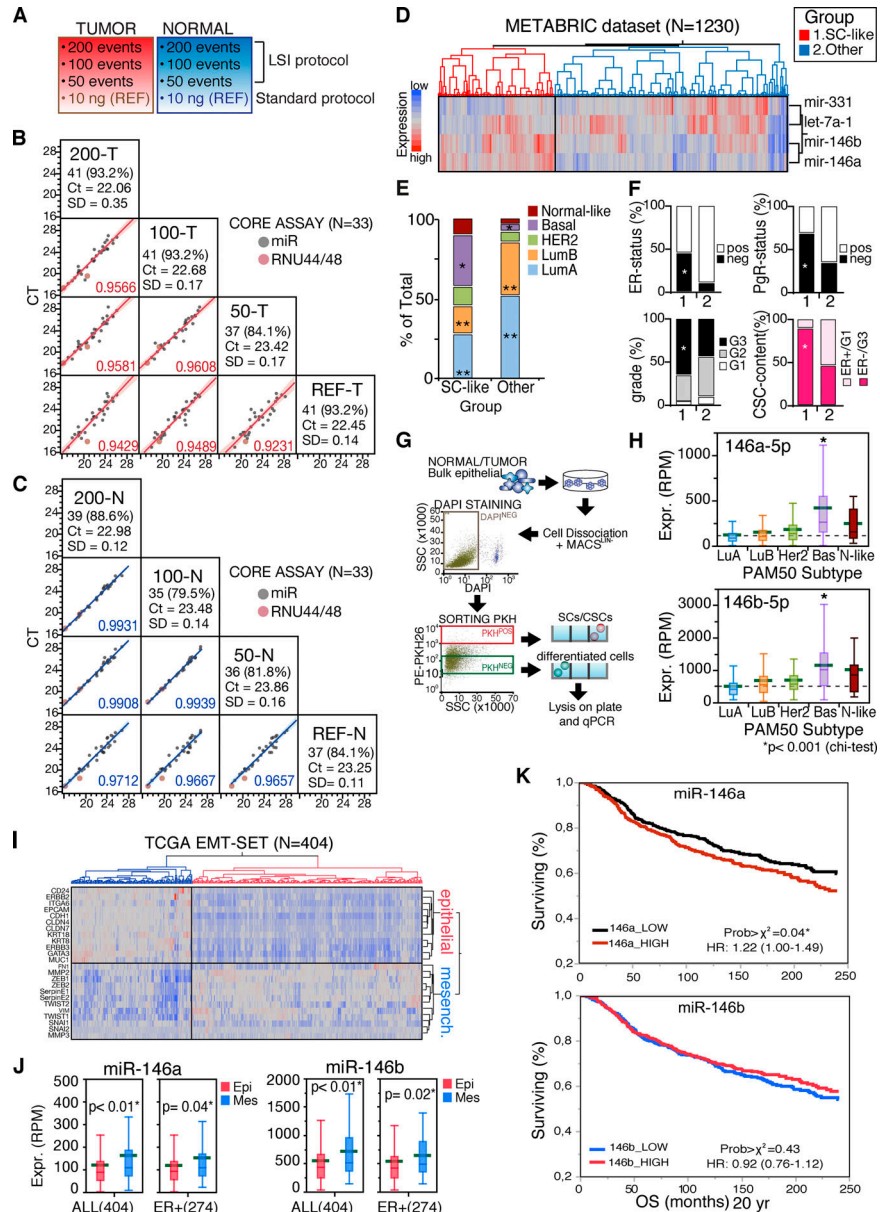

**Figure S1.  Generation of the LSI protocol and analysis of miR-146 levels in BC datasets. (A)** The LSI protocol was verified using 200, 100, and 50 FACS-sorted events, from one tumor and one normal primary sample, and compared with 10 ng of RNA from the same samples extracted with standard protocol. **(B and C)** Scatterplot matrix shows the expression of miRNAs (raw Ct) from LSI (200, 100, and 50 FACS-sorted events) or reference (10 ng) protocols. The percentage of detection with respect to the total is reported for 44 detected miRNAs (<30 Ct), with median Ct measured on two technical replicates ± SD. The linear correlation between the two protocols was determined using the core assay (33 miRNAs) and reported as a determination coefficient. RNU44/48 used as controls. Raw data are available in Table S1. **(D)** Hierarchical clustering (Ward method, with standardized data) of 1,230 primary breast tumors according to miR-146a/b, miR-331-3p, and Let-7a expression (METABRIC dataset). Two main clusters were identified. Data are reported as reads per million mapped (RPM). **(E)** Contingency analysis on clusters (identified in D), according to the PAM50 classification. Data are reported as percentage of the total number of samples. SC-like group (cluster 1) is significantly enriched of basal tumors (*, P < 0.001) and depleted of luminal tumors (**, P < 0.0001; $\chi^2$ Pearson coefficient). The group "Other" is depleted of basal (*, P < 0.001) and enriched of luminal tumors (**, P < 0.0001). **(F)** Contingency analysis as in E, according to hormone receptor status (ER and PgR), tumor grade, and CSC content. Data are reported as percentage of the total number of samples. The enrichments observed within the SC-like group are statistically significant (*, P < 0.001). **(G)** Scheme summarizing the workflow used for miRNA detection in PKH-labeled cells derived from mammospheres of primary cultures (see Materials and methods). **(H)** 886 primary tumors from the TCGA dataset, grouped according to the PAM50 classification, were stratified by miR-146a/b expression levels. Data are shown as box plots, mean RPM; P values calculated with Wilcoxon are also reported (*, P < 0.001). **(I and J)** Hierarchical clustering (Ward method, with standardized data) of 404 breast primary tumors from the TCGA dataset according to the expression of a tailor-made EMT signature. Two main clusters were identified, according to the expression of epithelial (Epi; in red) or mesenchymal markers (Mes; in blue). miR-146a/b expression was measured in each cluster identified in I. High miR-146a/b expression levels significantly correlated with mesenchymal features, independently from the molecular subtype. Data are reported as mean RPM; *, P value (each air, Student's t test). **(K)** Kaplan-Meier analysis of BC patients from METABRIC dataset (n = 1,217) according to miR-146a-b levels (hazard ratio [HR] and 95% CI were calculated in univariate analysis). Briefly, we selected from the METABRIC dataset 1,217 patients for whom clinical information was available on cBioportal. miR-146a-b levels were defined high or low, over the median value of expression in all the patients analyzed. OS, overall survival.

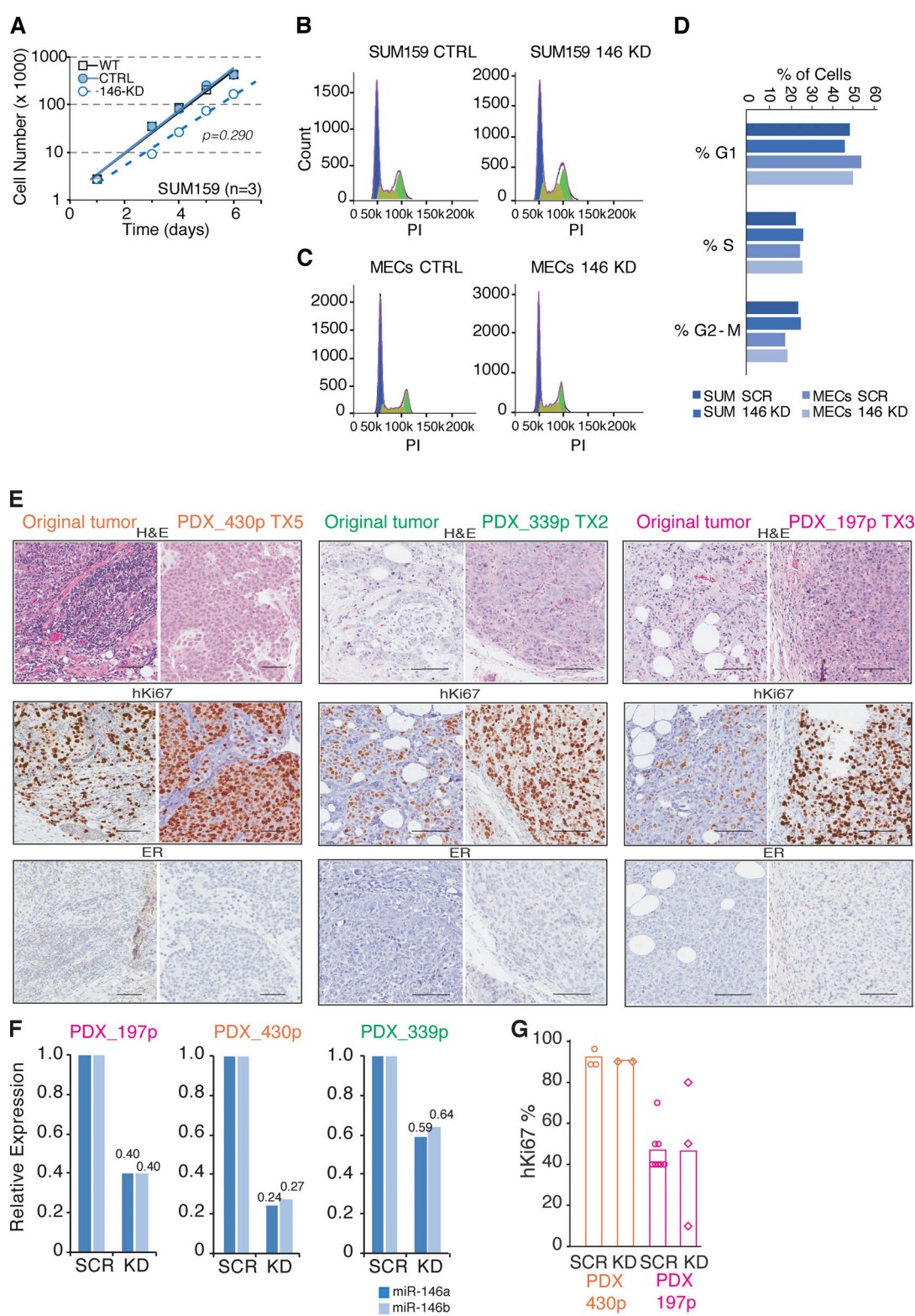

Figure S2. **Characterization of human BC PDXs and effect of miR-146 KD on proliferation. (A)** Growth curve of SUM159 cells infected with CTRL or miR-146 KD lentiviral vectors at each time point (24 h). Results are the mean of $n = 3$ biological replicas (P value by Student's $t$ test). **(B and C)** Cell cycle profile was analyzed in CTRL or miR-146 KD SUM159 cells (B) or murine MECs (C) by propidium iodide (PI) staining and FACS analysis. **(D)** Quantification of cell cycle phases from B and C. **(E)** Representative images of IHC analysis of the original tumor and its corresponding PDX, stained as indicated; scale bar, 100 µm. **(F)** Expression of miR-146 in purified tumors from human PDXs infected with CTRL or miR-146 KD lentiviral vector; data are reported as fold relative to CTRL, normalized on SNORD61 and SNORD72. **(G)** Quantification of hKi67 protein in two representative human PDXs following miR-146 KD. Each dot represents the percentage of hKi67-positive nuclei over the total cells in each slide analyzed by immunohistochemistry; mean value is reported as bar. In the PDX 339p, Ki67 staining could not be performed because of lack of material. ER, estrogen receptor; H&E, hematoxylin and eosin.

Tordonato et al.

The miR-146 family in normal and breast cancer stem cells

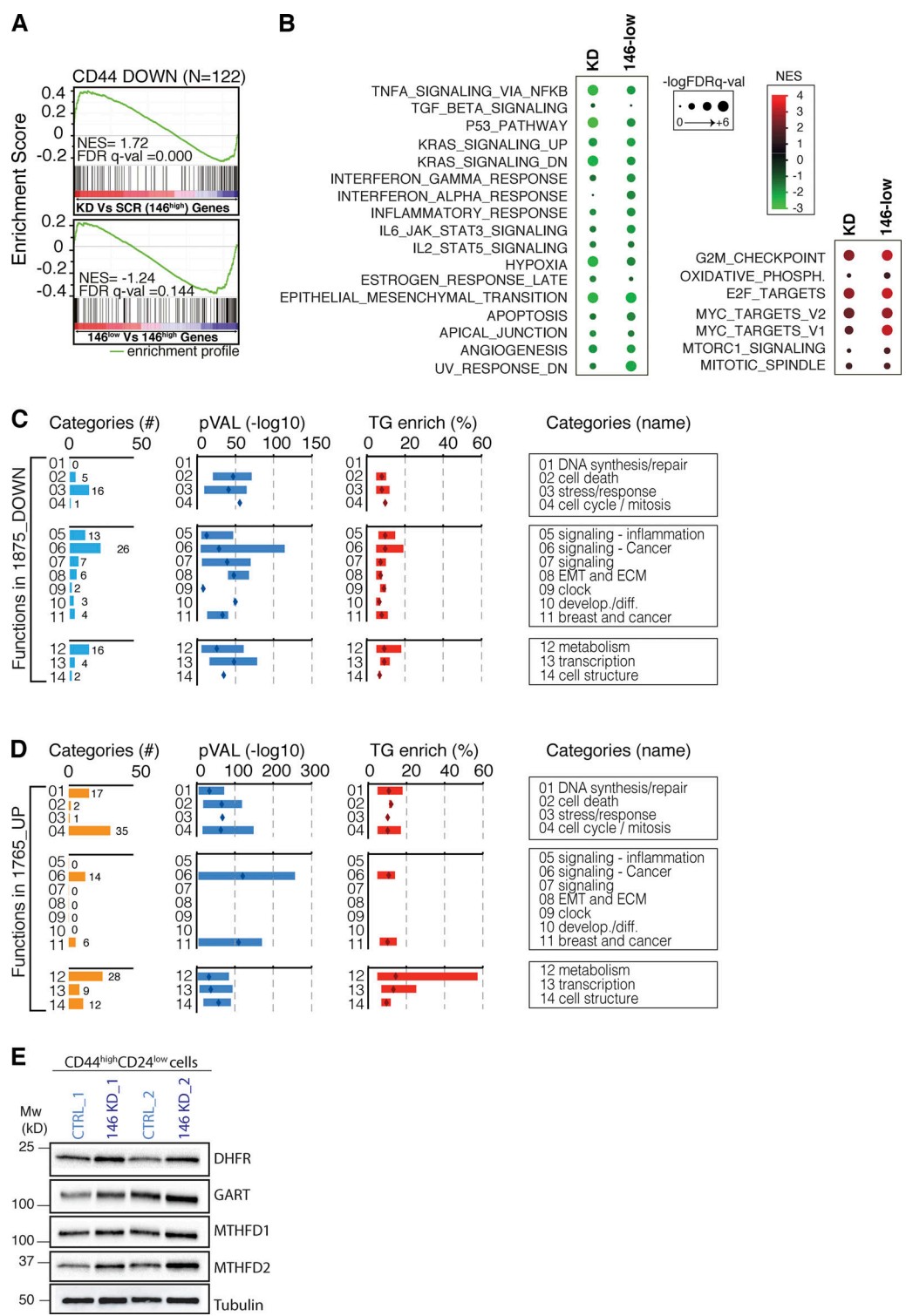

**Figure S3. Analysis of miR-146 transcriptional effects. (A)** Normalized enrichment score (NES) and FDR q-val show the association of the gene set down-regulated in CD44[high] cells (CD44 DOWN) with the gene expression changes in miR-146[low] or miR-146[high+KD] HMLE cells. **(B)** GSEA was used to correlate the gene expression changes observed in miR-146[low] or miR-146[high+KD] with the Molecular Signatures database (MSigDB v 6.2). The dot plot reports the NES and statistical significance (-Log$_{10}$FDR q-VAL) measured for each annotated gene set. **(C and D)** Enriched biological functions in 1875_DOWN (C) or in 1765_UP (D) genes identified in Fig. 5 G. The categories were retrieved by GSEA; those belonging to the same macro-area were grouped together and selected for significance (FDR q-Val < 0.01). Shown are the 13 macro-categories, with number of entries in yellow/light blue, P values in dark blue (bars for minimum-maximum values; diamonds for mean) and enrichment of miR-146 targets (TGs) in red (n = 945, as defined by TargetScan7.1; bars for minimum-maximum values; diamonds for mean). ECM, extracellular matrix; develop./diff., development/differentiation. **(E)** The blot shows the up-regulation of miR-146 targets belonging to the one-carbon metabolism pathway in CD44[high]/CD24[low] cells interfered for miR-146 or CTRL (two independent biological replicates); tubulin was used as loading control (left, molecular weight [MW] marker).

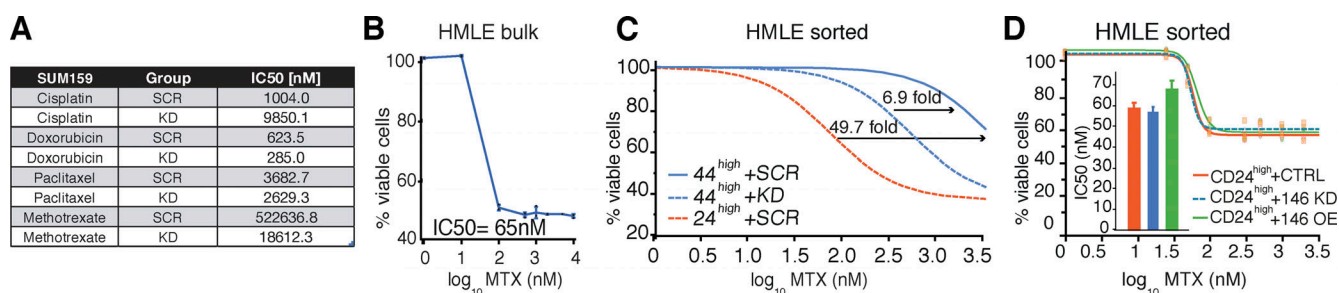

Figure S4. **Sensitivity to chemotherapy treatment upon miR-146 manipulation. (A)** IC50 concentrations were calculated by parallel fitting of the data shown in Fig. 6, A–D. For each drug, different numbers of biological replicates are shown, each performed in technical replicates. **(B)** Dose–response curve of HMLE WT cells treated with MTX for 72 h (n = 2). **(C)** Fitting of the curves shown in Fig. 6 E. The relative potency of MTX was calculated by fitting the data according to Logistic 4P script (JMP software) over control (CD44$^{high}$ + SCR). **(D)** Dose–response curve of HMLE CD44$^{low}$/CD24$^{high}$ in presence of miR-146 KD, miR-146 overexpression (OE), or CTRL, treated with MTX for 72 h. IC50 concentrations were estimated by parallel fit estimation (JMP software, n = 1 in technical triplicates).

**Four tables are provided online as separate Excel files. Table S1 shows raw data from the LSI protocol. Table S2 shows analysis of SC-specific miRNAs' signature in the TCGA dataset. Table S3 shows analysis of SC-specific miRNAs' signature in the METABRIC dataset. Table S4 shows a list of pathways regulated by miR-146 in SC-like cells.**

