## [Peer Review File · The Journal of Cell Biology]

miR-146 connects stem cell identity with metabolism and pharmacological resistance in breast cancer

Chiara Tordonato, Matteo Marzi, Giovanni Giangreco, Stefano Freddi, Paola Bonetti, Daniela Tosoni, Pier Paolo Di Fiore, and Francesco Nicassio

Corresponding Author(s): Francesco Nicassio, Italian Institute of Technology and Pier Paolo Di Fiore, IEO, Istituto Europeo di Oncologia

Review Timeline:	Submission Date:	2020-09-10
	Editorial Decision:	2020-11-02
	Revision Received:	2021-01-26
	Editorial Decision:	2021-02-19
	Revision Received:	2021-02-26

Monitoring Editor: Ira Mellman

Scientific Editor: Andrea Marat

Transaction Report:

DOI: <https://doi.org/10.1083/jcb.202009053>

November 2, 2020

Re: JCB manuscript #202009053

Dr. Francesco Nicassio
Italian Institute of Technology
Center for Genomic Science of IIT@SEMM
c/o Campus IFOM-IEO Via Adamello 16
Milan, Milan 20139
Italy

Dear Dr. Nicassio,

Thank you for submitting your manuscript entitled "miR-146 connects stem cell identity with metabolism and pharmacological resistance in breast cancer". The manuscript was assessed by expert reviewers, whose comments are appended to this letter. We invite you to submit a revision if you can address the reviewers' key concerns, as outlined here.

You will see that the reviewers are both positive regarding the potential impact of your paper, however have provided constructive comments to further improve your study. I agree that experiments with a second cell line would certainly be welcome, however if you do not find this necessary please provide a cogent explanation. Otherwise, please address all of the remaining reviewer comments in your revised manuscript.

GENERAL GUIDELINES:

Text limits: Character count for an Article is < 40,000, not including spaces. Count includes title page, abstract, introduction, results, discussion, acknowledgments, and figure legends. Count does not include materials and methods, references, tables, or supplemental legends.

Figures: Articles may have up to 10 main text figures. Figures must be prepared according to the policies outlined in our Instructions to Authors, under Data Presentation, <https://jcb.rupress.org/site/misc/ifora.xhtml>. All figures in accepted manuscripts will be screened prior to publication.

Supplemental information: There are strict limits on the allowable amount of supplemental data. Articles may have up to 5 supplemental figures. Up to 10 supplemental videos or flash animations are allowed. A summary of all supplemental material should appear at the end of the Materials and

methods section.

As you may know, the typical timeframe for revisions is three to four months. However, we at JCB realize that the implementation of social distancing and shelter in place measures that limit spread of COVID-19 also pose challenges to scientific researchers. Lab closures especially are preventing scientists from conducting experiments to further their research. Therefore, JCB has waived the revision time limit. We recommend that you reach out to the editors once your lab has reopened to decide on an appropriate time frame for resubmission. Please note that papers are generally considered through only one revision cycle, so any revised manuscript will likely be either accepted or rejected.

Thank you for this interesting contribution to Journal of Cell Biology. You can contact us at the journal office with any questions, cellbio@rockefeller.edu or call (212) 327-8588.

Sincerely,

Ira Mellman, Ph.D.
Editor

Andrea L. Marat, Ph.D.
Senior Scientific Editor

Journal of Cell Biology

Reviewer #1 (Comments to the Authors (Required)):

In this paper Tordonato and colleagues identify miR145 as critical determinant of CSC/stem cells self-renewal in breast cancer and postulated its connection with folate metabolism thus suggesting a vulnerability of breast cancer cells that could be exploited therapeutically.

The paper is of potential interest, however a few concerns remain to be addressed, find them below:

MiR-146a/b expression identifies cells with SC/CSC identity:

- "In BC, the proportion of cells with tumour-initiating ability (herein operationally equaled to CSCs) correlates with the molecular/biological characteristics of the tumour and its aggressiveness": do miR146a/b levels correlate with prognosis in TCGA or METABRIC data?

-Isolation and analysis of CSC is normally based on a combination of assays, here the authors used PKH26 incorporation. It would be nice to see in supplemental stainings for CD44 and CD24 to estimate the purity of CS isolation.

-(Minor comment) "The silencing of miR-146 in these PDXs reduced the frequency of TICs by 5-fold (Figure 3C-F), in the absence of significant effects on proliferation", correct but it would still be nice

to see a FISH for miR146 in PDX and/or a QPCR in FACS sorted CSC subpopulations.

- "miR-146 KD accelerated the conversion of the CD44^{high}/CD24^{low} subpopulation towards the original cell Population". The nature of the assay performed in figure 4N is not clear, it should be better explained in the text.

- "To gain mechanistic insights into miR-146 functions in the mammary SC compartment, we performed a comparative analysis of the transcriptomes": these experiments have been performed in duplicate thus making difficult a statistical evaluation of the results.

- Can you retrieve the miR146 seed in the upregulated set upon miR146 inhibition and if so in which bins?

MiR-146a/b regulates quiescence and one-carbon pool metabolism

- Can you identify direct targets proving this connection?

- Do you detect an inverse correlation between those targets and the miRNA in the PDX tissues upon miR-146 KD?

miR-146 depletion synergizes with MTX treatment

- Can the effects be reverted by addition of folic acid to the cells?

Reviewer #2 (Comments to the Authors (Required)):

In their study Tordonato et. al show that miR-146 is highly expressed in normal mammary stem cells (SCs) and in cancer stem cell (CSC) populations in breast tumors. The authors show that cell populations in normal mammary cell lines and in a breast cancer cell line, SUM159, that show high expression of miR-146 have a higher sphere forming efficiency (SFE) than cell populations from the same cell lines that show low expression of miR-146, arguing that miR-146 high cells have a higher capacity for self-renewal. Knock-down of miR-146 in these cells led to reduced SFE and fewer tumor initiating cells (TICs), arguing that miR-146 plays a functional role in mediating the ability of these cells to self-renew. Through comparative transcriptional analysis of miR-146-high and miR-146 low (endogenously low or low due to knock-down) cells, the authors identify pathways mediated by miR-146, including metabolic processes. Finally, the authors reason that miR-146 might mediate drug response and find that knock-down of miR-146 renders SUM159 cells sensitive to methotrexate but not to other common chemotherapeutics.

This study provides convincing evidence that miR-146 expression mediates SC/CSC phenotypes associated with self-renewal in normal mammary epithelial cells and in breast cancer cell lines. The observation that knock-down of miR-146 renders cells more sensitive to methotrexate treatment is interesting and, though it requires more extensive follow up, suggest therapies that target miR-146 could be effective in the clinic. Overall, the manuscript is well written and the experiments are straight-forward and well-controlled. However, the strength of the study would increase significantly if the authors were able to confirm the observations regarding methotrexate sensitivity in more than one tumor line.

Some specific points:

Figure 3 and related Figure S2:

1. In the TIC experiments using PDX lines, the number of mice used per line is inconsistent between

lines -- e.g. for the same dose, 4 for one line, 5 for another, and 8 for the last. The authors also do not use the same 3 doses for all 3 lines. While they use 25k, 10k, and 5k for 2 of the lines (430p and 339p), they use 50k, 25, and 5k for the third line (197p). These discrepancies should be corrected or explained.

2. The percent Ki67 quantification is shown for PDX_430p and PDX_197p but not for PDX_339p.

Figure 4:

3. The authors do not cite the reference for the microRNA sensor construct in the text (only in the methods).

4. The authors do not clearly describe the sensor construct in the text -- e.g. NGFR is used as a reporter for the presence of the construct and for normalization.

5. In N the baseline levels of CD44^{hi} cells for the population should be shown on the graph for reference.

Figure 5

6. The authors never mention what cell line they used for the transcriptional analysis of miR-146 high versus miR-146 low and miR-146 kd versus control cells. This needs to be clearly stated.

7. The labeling for this figure is confusing. Instead of "sensor" the authors should use a label that clearly indicates what the sample is -- miR-146 low cells.

Figure 6 and related Figure S4:

8. The observation that miR-146 kd renders xenografted tumor cells sensitive to methotrexate treatment was only performed in a single tumor cell line, SUM159. In figure 3, the authors use PDX lines in which miR-146 has been knocked-down. Does this knock-down also render these PDX lines more sensitive to methotrexate?

9. Does expression of miR-146 correlate with patient response to methotrexate?

10. The author's statement in reference figure S4D, is confusing -- "the non-SC population did not show any change in MTX sensitivity upon miR-146 manipulation (KD or overexpression)." This suggests a context specific role for miR-146, which should be discussed briefly in the discussion.

Reviewer #1:

(we numbered the reviewer's comments for clarity)

1. In BC, the proportion of cells with tumour-initiating ability (herein operationally equaled to CSCs) correlates with the molecular/biological characteristics of the tumour and its aggressiveness: do miR146a/b levels correlate with prognosis in TCGA or METABRIC data?

R. Agree. We tested this possibility in the METABRIC dataset, which contains better clinical information with respect to the TCGA dataset. The level of expression of miR-146a correlated with disease outcome (HR 1.22, $p = 0.04$), while no differences could be observed for miR-146b. These data are shown underneath and have been included in **Supplemental Figure S1K**.

Figure S1K: Kaplan Meier analysis of BC patients from MetabRIC dataset (N= 1217) according to miR-146a-b levels (HR and 95% CI were calculated in univariate analysis). Briefly, we selected from the MetabRIC dataset, 1217 patients for whom clinical informations were available on cBioportal. miR-146a-b levels were defined high or low, over the median value of expression in all the patients analyzed.

2. Isolation and analysis of CSC is normally based on a combination of assays, here the authors used PKH26 incorporation. It would be nice to see in supplemental staining for CD44 and CD24 to estimate the purity of CS isolation.

R. This is an interesting point. Stem cell (SC) isolation is based on approaches that exploit certain properties of SCs, which might vary as a function of the experimental setting. In particular, PKH staining is based on the co-segregation of SC and quiescence phenotypes, and it has been developed to exploit this property in *ex vivo* contexts such as organoid or mammosphere cultures (see (Pece et al., 2010)). Conversely, CD44/CD24 staining has been developed on primary tumors or in cultured cells, systems in which the SC phenotype segregates with the basal phenotype. Unfortunately, there is no proper context in which both can be used successfully (or readily substitute for each other). In mammospheres, where PKH is used, CD44/CD24 staining is not effective since cells cultured as mammosphere become basal-like and there is no further stratification by CD44^{high}/CD24^{low} staining. PKH cannot be used on fresh tumor samples for practical reasons, and neither on cultured cells, since there is no segregation of stemness with quiescence (these are transformed cells and they all duplicate, even the stem-like subpopulation). We can support this contention with a relevant case, shown in **Appendix A**, included here only for the reviewer's perusal.

In the figure below, we show single-cell transcriptional profiles of 16195 cells, in which we compared the HMLE bulk population (8450 cells) with the CD44^{high} SC-like population (7745 cells, isolated from HMLE cells by FACS sorting). Some conclusions can be drawn.

A. The single cell profiles reveal that the two cellular populations overlap to a limited extent, highlighting different subgroups (clusters) composed of CD44^{high}/CD24^{low} cells in blue as opposed to bulk cells in red.

B-C-D. Some of the CD44^{high}/CD24^{low} clusters (from CL1 to CL5) are enriched in the expression of genes related to breast SCs, thereby validating the approach (i.e. EMT genes, or SC-signatures from published works).

E-F. Using the expression of cell cycle genes, it is possible to estimate the cell cycle profile of each cells in each sample group (CD44 vs bulk) and in each of the clusters. No difference in cell cycle distribution exists between CD44^{high} stem-like population and HMLE bulk population. Therefore, PKH staining, which is based on quiescence, cannot be coupled with CD44/CD24 staining.

In conclusion, the best practice is to apply each of the two purification procedures (PKH or CD44-CD24 staining) to the contexts to which they are best suited. It is worth mentioning that miR-146 was observed consistently upregulated in different contexts and using each of the two approaches: i) in human BCSCs isolated from tumor biopsies using FACS sorting based on CD44^{high}CD24^{low} lineage⁻ (Figure 1F), from Shimono et al, Cell 2009 (PMID: 2731699); and in PKH⁺ cells isolated from human and mouse mammospheres (Figures 1B and 1F). Hence, miR-146 upregulation is consistent between the two approaches and can be considered a unifying element over the different properties used to purify SCs.

Appendix A: A-B. FACS sorted CD44^{high} or Bulk HMLE cells (**A**, left panel) were analyzed by single-cell transcriptomics (10X Chromium Single Cell 3' Solution V3.1). Raw sequencing data were processed through the Cell Ranger (v4.0) and Seurat (v3.0) tools in order to filter the dataset, define the dimensionality and perform clustering of the cells by non-linear dimensional reduction (tSNE). We analyzed 16195 single cell profiles (CD44,7745; Bulk, 8450; **A**, right panel), that were assigned to 10 different clusters (**B**). **C.** Proportion of CD44^{high} or Bulk cells within each cluster (% of the total number of cells). **D.** The bar

graph shows the Normalized Pathway Score for three gene signatures associated to normal breast SCs in each cluster. Shown are median centered Pathway Scores, as calculated by the ModuleScore function of Seurat3 using as input the following gene-sets: EMT-genes: Molecular Signature Database HALLMARK_EPITHELIAL_MESENCHYMAL_TRANSITION; STEM-Lim: Molecular Signature Database LIM_MAMMARY_STEM_CELL_UP; STEM-Pece: genes highly expressed in PKH^{POS} cells from Pece et al. Cell 2010 ("3/3 signature"). **E.** Distribution of cell cycle phases (G1, S, G2M) by Sample (CD44^{high}, Bulk). Each cell was assigned to a specific phase by the CellCycleScoring function of Seurat3. **F.** Distribution of cell cycle phases within each cluster. Each cell was assigned to a specific phase by the CellCycleScoring function of Seurat3.

3. (Minor comment) – ‘The silencing of miR-146 in these PDXs reduced the frequency of TICs by 5-fold (Figure 3C-F), in the absence of significant effects on proliferation’, correct but it would still be nice to see a FISH for miR146 in PDX and/or a QPCR in FACS sorted CSC subpopulations.

R. Agree. We tried to set up an ISH approach to measure the expression of 146 directly on primary mammary samples; unfortunately we did not obtain clear experimental data due to technical problems related to high background, poor sensitivity of the probes and the low basal expression of the two miRNAs. However, we could measure miR-146 levels in PDX upon silencing by using RT-QPCR. Results demonstrate the effective silencing of the two miRNAs in PDX tumors. These data are now included as **Supplemental Figure S2F** of the revised paper, and are also shown underneath.

4. ‘miR-146 KD accelerated the conversion of the CD44^{high}/CD24^{low} subpopulation towards the original cell Population’. The nature of the assay performed in figure 4N is not clear, it should be better explained in the text.

R. Agree. The manuscript has been revised accordingly (page **8**, lines **162-166** of the revised manuscript and page **25**, legend to **Figure 4N**).

5. ‘To gain mechanistic insights into miR-146 functions in the mammary SC compartment, we performed a comparative analysis of the transcriptomes’: these experiments have been performed in duplicate thus making difficult a statistical evaluation of the results.

R. We apologize for the lack of clarity in our original manuscript. We actually used four independent experiments, coming from two completely independent approaches, each performed in biological duplicates. We then used **all four experiments** with a ranking approach to select down- and upregulated genes in response to miR-146 KD, as depicted in *Figure 5G*. Having used a ranking approach that divides the transcriptome into groups, the statistical evaluation was made on a categorical basis using the chi-test, whose values are shown in the same *Figure 5G*. The high chi-square shows that it is very unlikely that genes are regulated by chance in four independent analyses.

6. Can you retrieve the miR146 seed in the upregulated set upon miR146 inhibition and if so in which bins?

R. We apologize for the lack of clarity in our original version; actually, these analyses were already present in the original paper. We looked for miRNA seed in the gene set regulated by miR146 inhibition. To this end, we used Targetscan, which estimates the probability of a gene of being a target by looking at the type of ‘seed’ and the overall sequence context, measured as ‘context score’. The lower the score the higher is the probability of being a target. As shown in the panel below, genes predicted with high and medium probability (<-0.30 and <-0.15 context score) were effectively upregulated (**A**). These genes

were considered as the predicted targets for miR-146 (945 genes, 146 TG) and are reported in *Figure 5B-C*, as effectively upregulated in miR-146 low cells (miR-low) and upon miR-146 silencing (KD). In addition, we found significant enrichment of predicted targets in the first bins (bins 2-3-4, upregulated genes, **B**) as shown in *Figure 5F*. These genes were used to select for direct miRNA targets as shown in *Figure 5G-H*. In the panel below, we have represented the same data, in a different manner. We believe that our original descriptions are sufficient; however, we would be ready to add the panel below as a Supplemental Figure, if the reviewer thinks that this would help.

A. Genes in the two datasets analyzed (miR-low and miR-146 KD, expressed in \log_2 fold change vs. Ctrl) were divided in 5 classes according to the context score derived from Targetscan prediction ($-1.74 < \text{high} < -0.3$, $-0.3 < \text{medium} < -0.15$, $-0.15 < \text{low} < -0.1$ and $-0.1 < \text{very low} < -0.01$). Plotted are the cumulative distributions of mRNA fold changes, comparing the five different classes: the higher the context score, the stronger is the upregulation of the mRNAs with respect to the non-target category.

B. The ranked list of genes in miR-146low or 146high+KD (from the most upregulated to the most repressed) were divided in bins of ≈ 1100 elements/bin. The heatmap displays, in each bin, the enrichment score of the prediction classes defined in A. MiR-146 predicted targets ($n=945$, 'high' or 'medium' context score groups) were enriched in the first 4 bins.

7. MiR-146a/b regulates quiescence and one-carbon pool metabolism. Can you identify direct targets proving this connection?

R. We thank the reviewer for the comment, and we apologize for not having clearly explained this point in the original version of the paper. Indeed, panel 5G shows the **direct targets** of miR-146 found in SC-like cells. The targets related to one carbon pool metabolism are those highlighted in red in the figure, i.e. DHFR, GART, MTHFD1, and MTHFD2. In addition, we have also verified the effects at the protein level, through a WB analysis performed on the SC-like cells (CD44 high) subpopulation of HMLE cells, in which miR-146 was silenced. These data (also shown below) are now reported in **Supplemental Figure S3E** and further confirm the effects of miR-146 on enzymes of the one-carbon pool metabolism.

8. Do you detect an inverse correlation between those targets and the miRNA in the PDX tissues upon miR-146 KD?

R. We thank the reviewer for the comment. To answer the question, we extracted RNA from the PDX tumors (obtained with or without miR-146 silencing), and looked at the levels of target genes of the one-carbon pool (GART, MTHFD1, MTHFD2 genes) and well-known 146 targets (TRAF6, a positive control) by RT-qPCR. However, we couldn't find any anticorrelation for either the one-carbon pool genes **or the positive control**. There are several possible explanations for this negative result. Probably in the PDX tumors the effects are masked by the heterogeneity of the tumor population in which only a small number of CSCs is present vs. the bulk non-SC population (progenitors and differentiated cells). In these latter cells, obviously, one cannot expect any effect of miR-146 silencing, since the miRNA is expressed at very low levels. Indeed, miRNA effects on transcripts are clearly visible only in very controlled setting, by limiting the sources of transcriptional variability, and at early time points after miRNA manipulation to avoid indirect transcriptional effects.

9. miR-146 depletion synergizes with MTX treatment. Can the effects be reverted by addition of folic acid to the cells?

R. This is an interesting point and we agree that it would be interesting to investigate whether folic acid supplementation could reverse some of the effects of miR-146. As explained in the letter to the Editor, we would need more than a year to address this point, for regulatory reasons. Therefore, we believe that such investigation should be pursued in the context of follow-up studies.

Reviewer #2

Figure 3 and related Figure S2:

1. In the TIC experiments using PDX lines, the number of mice used per line is inconsistent between lines -- e.g. for the same dose, 4 for one line, 5 for another, and 8 for the last. The authors also do not use the same 3 doses for all 3 lines. While they use 25k, 10k, and 5k for 2 of the lines (430p and 339p), they use 50k, 25, and 5k for the third line (197p). These discrepancies should be corrected or explained.

R. We thank the reviewer for the comment. There is a technical reason related to these differences, mostly due to the fact that each PDX has different growth properties, and therefore the latency for tumor growth and the number of cells that can be purified at each passage from each single tumor are different. However, the results are correct, as we compared control and KD cells at the same dose in each of the PDXs and we derived the TIC frequency using 3 different doses. To prove that results are valid, we also re-calculated the TIC-frequency taking in consideration only the 25k and 5k doses, which are in common for all the PDXs. As shown in the panel underneath the differences between KD and CTRL are maintained with a statistical significance for all the PDXs.

		PDX_430p		PDX_339p		PDX_197p	
		Dose	Response	Dose	Response	Dose	Response
CTRL	25000	4/4	25000	5/5	25000	8/8	
	5000	2/2	5000	7/8	5000	3/9	
	TIC-frequency (upper/lower)	1 (1:1-1:13355)	TIC-frequency (upper/lower)	1:2404 (1:997-1:5797)	TIC-frequency (upper/lower)	1:8517 (1:4212-1:17221)	
146 KD	25000	2/4	25000	3/5	25000	3/12	
	5000	0/3	5000	4/8	5000	0/10	
	TIC-frequency (upper/lower)	1:43818* (1:11019-1:174247)	TIC-frequency (upper/lower)	1:15311* (1:6611-1:35462)	TIC-frequency (upper/lower)	1:103665* (1:33621-1:319632)	
		* p=0.002		* p=0.002		* p=4.2e-05	

2. The percent KI67 quantification is shown for PDX_430p and PDX_197p but not for PDX_339p.

R. We thank the reviewer for the comment. Unfortunately, we were unable to perform Ki67 quantification on the histological material for PDX_339p. This particular PDX has a slow growth kinetics and typically has few epithelial cells. Most of the material was used to check for miR-146 KD by RT-QPCR, leaving a

low amount of material included as FFPE for histological analyses. Unfortunately with such material we were not able to quantify the Ki67 levels properly. To acknowledge this, we have amended the manuscript as follows “when we measured proliferation effects by Ki67 staining (which was possible in in two out of three PDXs because of availability of material), we did not score differences in KD vs. SCR. (page 7, lines 139-142 of the revised manuscript).

Figure 4:

3. The authors do not cite the reference for the microRNA sensor construct in the text (only in the methods).

R. Agree. We have amended the manuscript with the appropriate reference (page 7, line 149 of the revised manuscript).

4. The authors do not clearly describe the sensor construct in the text -- e.g. NGFR is used as a reporter for the presence of the construct and for normalization.

R. Agree. We have described more clearly the feature of the sensor in the manuscript (page 7-8, lines 145-149 of the revised manuscript).

5. In N the baseline levels of CD44hi cells for the population should be shown on the graph for reference.

R. Agree. Baseline levels of CD44hi/CD24lo population are 95% for both SCR and 146-KD cells. This information has been added in **legend to Figure 4N**, and it is also reported underneath.

CD44 h/CD24l	day 0	day 3	day 5	day 7
HMLE SCR	95	71.8	71.75	57.3
HMLE 146 KD	95	35.95	26.265	9.895

Figure 5

6. The authors never mention what cell line they used for the transcriptional analysis of miR-146 high versus miR-146 low and miR-146 kd versus control cells. This needs to be clearly stated.

R. Agree. The cells used are HMLE, we added this information to **Figure 5A** of the revised manuscript.

7. The labeling for this figure is confusing. Instead of "sensor" the authors should use a label that clearly indicates what the sample is -- miR-146 low cells.

R. Agree. We have replaced the term 'sensor' with '146-low' (as compared to 146-high) in **Figure 5, panels B, C, E, F, G and H**.

Figure 6 and related Figure S4:

8. The observation that miR-146 kd renders xenografted tumor cells sensitive to methotrexate treatment was only performed in a single tumor cell line, SUM159. In figure 3, the authors use PDX lines in which miR-146 has been knocked-down. Does this knock-down also render these PDX lines more sensitive to methotrexate?

R. We agree that it would be interesting to further confirm these effects using PDX as well, as suggested by the reviewer. However, as explained in the letter to the Editors (and also following their suggestions), we would need more than a year to address this point, for law regulatory reasons. Therefore, we believe that such investigation should be pursued in the context of follow-up studies

9. Does expression of miR-146 correlate with patient response to methotrexate?

R. We agree with the reviewer that this would be an interesting analysis. However, in breast cancer, MTX is not generally used either as first line of therapy, or as monotherapy in relapsed patients; it is usually associated with other drugs, typically 5-FU (Yang et al., 2020). Thus, we could not identify any suitable patient cohort to perform the experiment.

10. The author's statement in reference figure S4D, is confusing -- "the non-SC population did not show any change in MTX sensitivity upon miR-146 manipulation (KD or overexpression)." This suggests a context specific role for miR-146, which should be discussed briefly in the discussion.

R. Agree. The sum of our evidence suggests a role for miR-146 in the SC population, rather than in all cancer cells. To acknowledge this, we have amended the manuscript as follows: "In addition, drug sensitivity was affected only and specifically in the SC-like population, while no major effects were observed in the non-SC population (Figure S4D), which further suggests a context-specific role for miR-146 in the breast stem cells rather than a more general effect on bulk epithelial cells." (page **15**, lines **317-320** of the revised manuscript).

References:

- Pece, S., D. Tosoni, S. Confalonieri, G. Mazzarol, M. Vecchi, S. Ronzoni, L. Bernard, G. Viale, P.G. Pelicci, and P.P. Di Fiore. 2010. Biological and molecular heterogeneity of breast cancers correlates with their cancer stem cell content. *Cell*. 140:62-73.
- Yang, V., M.J. Gouveia, J. Santos, B. Kokschi, I. Amorim, F. Gartner, and N. Vale. 2020. Breast cancer: insights in disease and influence of drug methotrexate. *Rsc Med Chem*. 11:646-664.

February 19, 2021

RE: JCB Manuscript #202009053R

Dr. Francesco Nicassio
Italian Institute of Technology
Center for Genomic Science of IIT@SEMM
c/o Campus IFOM-IEO Via Adamello 16
Milan, Milan 20139
Italy

Dear Dr. Nicassio:

Thank you for submitting your revised manuscript entitled "miR-146 connects stem cell identity with metabolism and pharmacological resistance in breast cancer". We would be happy to publish your paper in JCB pending final revisions necessary to meet our formatting guidelines (see details below).

A. MANUSCRIPT ORGANIZATION AND FORMATTING:

Full guidelines are available on our Instructions for Authors page, <https://jcb.rupress.org/submission-guidelines#revised>. **Submission of a paper that does not conform to JCB guidelines will delay the acceptance of your manuscript.**

1) Text limits: Character count for Articles is < 40,000, not including spaces. Count includes title page, abstract, introduction, results, discussion, acknowledgments, and figure legends. Count does not include materials and methods, references, tables, or supplemental legends.

2) Figures limits: Articles may have up to 10 main text figures.

3) * Figure formatting: Scale bars must be present on all microscopy images, including inset magnifications. Molecular weight or nucleic acid size markers must be included on all gel electrophoresis. *

4) Statistical analysis: Error bars on graphic representations of numerical data must be clearly described in the figure legend. The number of independent data points (n) represented in a graph must be indicated in the legend. Statistical methods should be explained in full in the materials and methods. For figures presenting pooled data the statistical measure should be defined in the figure legends. Please also be sure to indicate the statistical tests used in each of your experiments (either in the figure legend itself or in a separate methods section) as well as the parameters of the test (for example, if you ran a t-test, please indicate if it was one- or two-sided, etc.). Also, if you used parametric tests, please indicate if the data distribution was tested for normality (and if so, how). If not, you must state something to the effect that "Data distribution was assumed to be normal but this was not formally tested."

- 5) Abstract and title: The abstract should be no longer than 160 words and should communicate the significance of the paper for a general audience. The title should be less than 100 characters including spaces. Make the title concise but accessible to a general readership.
- 6) Materials and methods: Should be comprehensive and not simply reference a previous publication for details on how an experiment was performed. Please provide full descriptions in the text for readers who may not have access to referenced manuscripts.
- 7) Please be sure to provide the sequences for all of your primers/oligos and RNAi constructs in the materials and methods. You must also indicate in the methods the source, species, and catalog numbers (where appropriate) for all of your antibodies. Please also indicate the acquisition and quantification methods for immunoblotting/western blots.
- 8) Microscope image acquisition: The following information must be provided about the acquisition and processing of images:
 - a. Make and model of microscope
 - b. Type, magnification, and numerical aperture of the objective lenses
 - c. Temperature
 - d. Imaging medium
 - e. Fluorochromes
 - f. Camera make and model
 - g. Acquisition software
 - h. Any software used for image processing subsequent to data acquisition. Please include details and types of operations involved (e.g., type of deconvolution, 3D reconstitutions, surface or volume rendering, gamma adjustments, etc.).
- 9) References: There is no limit to the number of references cited in a manuscript. References should be cited parenthetically in the text by author and year of publication. Abbreviate the names of journals according to PubMed.
- 10) Supplemental materials: There are strict limits on the allowable amount of supplemental data. Articles/Tools may have up to 5 supplemental display items (figures and tables). Please also note that tables, like figures, should be provided as individual, editable files. A summary of all supplemental material should appear at the end of the Materials and methods section.
- 11) eTOC summary: A ~40-50-word summary that describes the context and significance of the findings for a general readership should be included on the title page. The statement should be written in the present tense and refer to the work in the third person.
- 12) Conflict of interest statement: JCB requires inclusion of a statement in the acknowledgements regarding competing financial interests. If no competing financial interests exist, please include the following statement: "The authors declare no competing financial interests." If competing interests are declared, please follow your statement of these competing interests with the following statement: "The authors declare no further competing financial interests."
- 13) ORCID IDs: ORCID IDs are unique identifiers allowing researchers to create a record of their various scholarly contributions in a single place. At resubmission of your final files, please consider providing an ORCID ID for as many contributing authors as possible.

B. FINAL FILES:

-- High-resolution figure and video files: See our detailed guidelines for preparing your production-ready images, <https://jcb.rupress.org/fig-vid-guidelines>.

****It is JCB policy that if requested, original data images must be made available to the editors. Failure to provide original images upon request will result in unavoidable delays in publication. Please ensure that you have access to all original data images prior to final submission.****

****The license to publish form must be signed before your manuscript can be sent to production. A link to the electronic license to publish form will be sent to the corresponding author only. Please take a moment to check your funder requirements before choosing the appropriate license.****

Thank you for this interesting contribution, we look forward to publishing your paper in Journal of Cell Biology.

Sincerely,

Ira Mellman, Ph.D.
Editor

Andrea L. Marat, Ph.D.
Senior Scientific Editor

Journal of Cell Biology

Reviewer #1 (Comments to the Authors (Required)):

The authors have now addressed all my concerns except for the synergism with MTX that was also a point raised by reviewer nr.2. In this case the title of the manuscript should be reevaluated.

Reviewer #2 (Comments to the Authors (Required)):

The authors have addressed all the reviewer comments. The manuscript is sufficiently developed for publication in JCB.